# EchoAttention: Exploiting Token-Pair Redundancy and Frame-Block Similarity for Efficient Video Generation

**Yifei Xia** [1 2]  **Fangcheng Fu** [3]  **Hao Yuan** [1]  **Suhan Ling** [1 2]  **Xupeng Miao** [1]  **Huixia Li** [2]  **Yuxi Ren** [2]  **Xin Xia** [2]  **Xuefeng Xiao** [2]  **Bin Cui** [1 4]

## Abstract

Diffusion Transformers (DiTs) are increasingly adopted for video generation, yet inference is dominated by the quadratic cost of 3D full attention. Sparse attention mitigates this bottleneck by exploiting *token-pair redundancy* and pruning query–key interactions. Nevertheless, its effectiveness on video generation is often constrained by non-sparse attention heads, making it hard to strike a good balance between inference speed and generation quality. To address this, we identify another pervasive but overlooked redundancy specific to video DiTs: *frame-block similarity*, where frame-blocks in attention weights exhibit highly similar distributions and can be well approximated by lightweight linear calibration. Motivated by this observation, we propose **EchoAttention**, which jointly leverages *token-pair redundancy* (*Sparse* operator) and *frame-block similarity* (*Echo* operator), together with a fine-grained routing policy learned via three-stage distillation. This design enables efficient handling of both sparse and non-sparse heads, overcoming the inherent ceiling of purely sparse attention and yielding a better speed–quality trade-off. Across public video DiTs, EchoAttention consistently improves the speed–quality frontier over SOTA sparse-attention baselines, reducing end-to-end latency up to 2.42× with minimal quality loss.

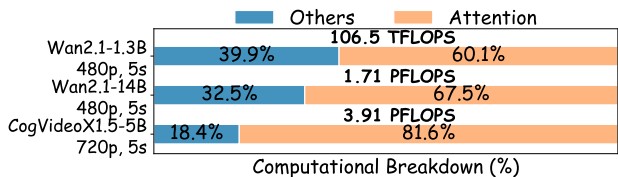

*Figure 1.* Attention breakdown under typical generation setups.

## 1. Introduction

**Background.** In recent years, Diffusion Transformers (DiTs) (Peebles & Xie, 2023) have demonstrated remarkable capability in video generation (Ho et al., 2022; Wan et al., 2025; Yang et al.; Kong et al., 2024; Jiang et al., 2024b; Zhao et al., 2025). However, as illustrated in Figure 1, generating high-resolution or longer videos remains compute-intensive (Xia et al., 2025), largely due to the quadratic cost of *3D full attention* (Vaswani et al., 2017; Yang et al.; Wu et al., 2025), which rapidly dominates runtime and limits the practicality of DiT models in video generation scenarios (Xia et al., 2025; Zhang et al.). Therefore, improving the efficiency of attention computation is paramount to accelerating video generation, especially at larger spatial or temporal scales (Xia et al., 2025).

**Current Approach.** A common approach is *sparse attention* (Child, 2019; Beltagy et al., 2020), which utilizes *token-pair redundancy* in attention weights by pruning query–key token pairs. As shown in the bottom middle of Figure 2, it reduces the quadratic cost by restricting attention to a subset of salient query–key pairs and masking out the rest. The retention ratio (equivalently, top-$k$) parameterizes the compute budget and, consequently, the resulting quality (Zhang et al., 2025c;b). For example, static methods such as SVG (Xi et al.), VSA (Zhang et al., 2025c), and SLA (Zhang et al., 2025a) fix top-$k$ to maximize fidelity under a predetermined compute budget, whereas dynamic methods such as SVG2 (Yang et al., 2025b) and SpargeAttn (Zhang et al., 2025b) adapt top-$k$ dynamically to satisfy a target quality constraint while minimizing computation.

**Limitation.** Despite their promise, sparse attention is ultimately limited by a class of *non-sparse heads* (Chen et al.,

[1]School of Computer Science & Beijing Key Laboratory of Software and Hardware Cooperative Artificial Intelligence Systems, Peking University [2]ByteDance Seed [3]School of Artificial Intelligence, Shanghai Jiao Tong University [4]Institute of Computational Social Science, Peking University (Qingdao). Correspondence to: Fangcheng Fu <ccchengff@sjtu.edu.cn>, Bin Cui <bin.cui@pku.edu.cn>.

*Proceedings of the 43rd International Conference on Machine Learning*, Seoul, South Korea. PMLR 306, 2026. Copyright 2026 by the author(s).

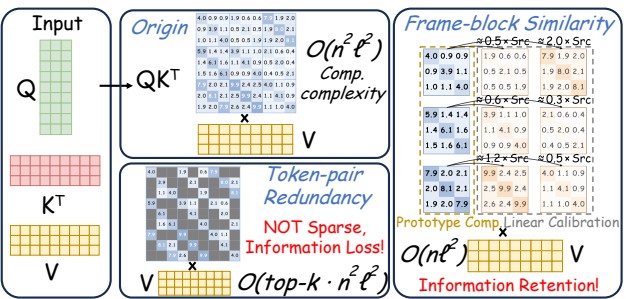

Figure 2. Difference between *token-pair redundancy* and *frame-block similarity*. Bottom middle: *sparse attention* leverages *token-pair redundancy* by pruning query–key pairs, but discards substantial mass on non-sparse heads. Right: a toy *frame-block similarity* example ($n=3, \ell=3$), where blocks within each frame-block row have similar distributions, so one block per row suffices, and the rest are well approximated by a simple linear calibration.

2025a; Wen et al., 2025), which dominate the speed–quality trade-off. We find strong head-wise heterogeneity in video DiTs (§3): many heads are naturally sparse, with attention mass captured by a small top-$k$, making them easy to sparsify. In contrast, a substantial fraction are *non-sparse*: their recall increases nearly linearly with top-$k$, indicating that salient mass is not concentrated on a small set of token pairs (Figure 4, §3). Maintaining quality on such heads requires a much larger top-$k$, leading to a high compute floor (Zhou et al., 2025); as a result, sparse methods either saturate in speedup or incur clear quality loss. Recently, SLA (Zhang et al., 2025a) augments sparsification with a *sparsity-first* plus *linear compensation* design, but it still remains constrained on non-sparse heads that are neither highly sparsifiable nor well captured by such simple compensation, leaving the bottleneck intact (§5.2). Overall, in video generation, prior sparse attention methods focus only on conventional *token-pair redundancy* within attention and thus often struggle to overcome the *non-sparse heads* bottleneck, ultimately capping the achievable gains.

**Key Observation.** Fortunately, beyond the widely exploited *token-pair redundancy*, we identify a pervasive yet largely overlooked *frame-block similarity* in attention weights of video DiTs that provides an orthogonal handle for accelerating *non-sparse heads*. Consider a video with $n$ latent frames and $\ell$ tokens per latent frame. Its attention weights naturally decompose into an $n \times n$ grid of $\ell \times \ell$ *frame-blocks* (§2.1). We observe strong distributional similarity among frame-blocks within the same block-row/column, and the residual differences are often well-approximated by a lightweight linear calibration (diagonal scaling) (§3). The right of Figure 2 gives a toy example where blocks within each frame-block row exhibit highly aligned attention-weight distributions. This suggests computing a single *prototype* block per row/col and recovering the remaining blocks via lightweight linear calibration, reducing the dominant atten-

tion cost from $O(n^2\ell^2 d)$ to $O(n\ell^2 d)$. In a word, although these non-sparse heads offer little *token-pair redundancy*, they exhibit strong similarity at the frame-block scale: attention across frames follows a shared *prototype* with only lightweight, correctable variations. This *frame-block similarity* motivates a complementary alternative to token-pair sparsification for non-sparse yet high-impact heads: *prototype reuse* with *lightweight linear calibration*.

**Challenges.** However, leveraging this similarity raises two key challenges. First, how can we convert *frame-block similarity* into real complexity reduction without computing dense attention? A naive route, explicitly measuring and fitting similarity (e.g., least squares), still requires materializing dense attention, costing $O(n^2\ell^2 d)$. The challenge is to identify representative prototypes and capture cross-frame variations at negligible overhead. Second, video DiTs are heterogeneous across heads (§3): some favor token-pair sparsification, while others are non-sparse yet well matched to our frame-block reuse. Therefore, automatically making fine-grained strategy selections to optimize the speed–quality frontier is a key challenge.

**Our Solution.** To address these challenges, we propose **EchoAttention**, a structured attention acceleration framework for video generation, which exploits both *token-pair redundancy* and *frame-block similarity*. EchoAttention instantiates a dual-operator system with automatic selection: a *Sparse* operator for sparse heads and an *Echo* operator tailored to non-sparse heads. The *Echo* operator further specializes into *Echo-Row* and *Echo-Col*, leveraging structured row/column similarity to turn redundant cross-frame attention into *prototype reuse* with learnable, lightweight linear calibration, reducing the dominant complexity from $O(n^2\ell^2 d)$ to $O(n\ell^2 d)$. Meanwhile, we adopt a three-stage distillation schedule that stabilizes training and learns an appropriate operator routing policy at *timestep–layer–head* granularity. Unlike prior sparse methods that rely on a unified sparse mechanism, EchoAttention does not force all heads into sparse attention. Instead, it preferentially exploits frame-block similarity on non-sparse, high-impact heads, achieving both speed and fidelity and breaking the bottleneck of conventional sparse attention.

Our technical contributions are summarized below.

- We quantify head-level *sparsifiability* heterogeneity and characterize the pervasive *frame-block similarity* in video DiTs.
- We propose **EchoAttention**, a dual-operator attention architecture that complements sparse attention (*Sparse* operator) with a low-complexity *Echo* operator utilizing *frame-block similarity* via prototype reuse and lightweight linear calibration.
- We introduce a three-stage distillation and fine-grained routing framework that learns automatic operator rout-

*Table 1.* Frequently used notation.

| Notation | Description |
|---|---|
| $n, \ell$ | Number of latent frames and tokens per latent frame |
| $L$ | Total latent video token length, $L = n\ell$ |
| $H, d$ | Number of attention heads and head dimension |
| $T, t$ | Number of denoising timesteps and one timestep |
| $\mathbf{W}, \mathbf{O}$ | Attention weights and Attention outputs |
| $\mathbf{W}_{ij}$ | Frame-Block of $\mathbf{W}$, $\mathbf{W}_{ij} \in \mathbb{R}^{\ell \times \ell}$ |
| $\mathbf{M}$ | Sparse mask (Sparse pattern), $\mathbf{M} \in \{0, 1\}^{L \times L}$ |
| top-$k$ | Retention ratio of $\mathbf{M}$, indicating the sparsity level (lower means sparser) |
| $\mathbf{B}, \mathbf{E}$ | Prototype frame-block and diagonal matrix used for linear calibration |

ing and stabilizes training.

- We implement efficient kernels for EchoAttention and demonstrate consistent end-to-end gains on multiple public video diffusion models.

**Conflict of Interest Disclosure.** Several authors are affiliated with ByteDance Seed. This work evaluates publicly available video generation models and sparse-attention baselines. To the best of our knowledge, none of the evaluated systems are ByteDance products. The authors declare no other financial conflicts of interest.

## 2. Background and Related Work

This section introduces the preliminary and related work of this paper. Table 1 summarizes frequently used notation.

### 2.1. 3D Full Attention and Frame-block Pattern

In 3D full attention (Vaswani et al., 2017) for video DiTs (Yang et al.; Gao et al., 2025; Li & Ge, 2025), the input latent video tokens can be regarded as a frame-wise sequence of length $L = n\ell$, where $n$ is the number of frames, and each frame contributes $\ell$ tokens. For each head, tokens are projected into query, key, and value matrices $\mathbf{Q}, \mathbf{K}, \mathbf{V} \in \mathbb{R}^{L \times d}$. The attention weights and outputs of this head are then computed as

$$\mathbf{W} = \mathrm{softmax}\left(\frac{\mathbf{Q}\mathbf{K}^\top}{\sqrt{d}}\right) \in \mathbb{R}^{L \times L}, \quad \mathbf{O} = \mathbf{W}\mathbf{V} \in \mathbb{R}^{L \times d}. \quad (1)$$

Under the frame-wise ordering $L = n\ell$, the attention weights $\mathbf{W}$ can be viewed as an $n \times n$ block matrix with frame-blocks $\mathbf{W}_{ij} \in \mathbb{R}^{\ell \times \ell}$:

$$\mathbf{W} = \begin{bmatrix} \mathbf{W}_{00} & \cdots & \mathbf{W}_{0(n-1)} \\ \vdots & \ddots & \vdots \\ \mathbf{W}_{(n-1)0} & \cdots & \mathbf{W}_{(n-1)(n-1)} \end{bmatrix}. \quad (2)$$

Equivalently, the cost of full attention is $O(n^2\ell^2 d)$. The quadratic dependence on both $n$ and $\ell$ still makes long sequences expensive.

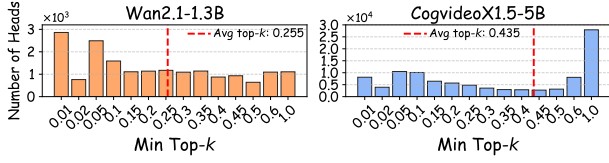

*Figure 3.* Head sparsity distribution. The y-axis counts heads by the minimum top-$k$ required to reach recall = 0.8. Smaller min top-$k$ indicates higher sparsity.

### 2.2. Sparse Attention

Sparse attention (Gao et al., 2024; Han et al., 2024; Xiao et al.; Munkhdalai et al.; Jiang et al., 2024a; Acharya et al.) reduces the quadratic cost of full attention by limiting which token pairs are allowed to interact. Formally, given query $\mathbf{Q} \in \mathbb{R}^{L \times d}$, key $\mathbf{K} \in \mathbb{R}^{L \times d}$, and a binary mask $\mathbf{M} \in \{0, 1\}^{L \times L}$ with top-$k$ specifying the admissible connections, where $\mathbf{M}[i, j] = 0$ means that the contribution from token $j$ to token $i$ is suppressed. The corresponding sparse attention output is

$$\mathbf{O}_{\mathrm{sparse}} = \mathrm{softmax}\left(\frac{\mathbf{Q}\mathbf{K}^\top}{\sqrt{d}} + \log \mathbf{M}\right)\mathbf{V}, \quad (3)$$

$\log \mathbf{M}$ is element-wise with $\log 1 = 0$ and $\log 0 = -\infty$.

In practice, many sparsity patterns (Yuan et al., 2024; Sun et al., 2025; Shen et al., 2025; Gu et al., 2025; Chen et al., 2025b) have been explored, such as purely sliding windows (Han et al., 2024), dilated or strided layouts, and mixtures of local and a few global tokens (Zaheer et al., 2020). Among these, *block-based* pattern (Zaheer et al., 2020) is widely adopted due to its regular memory access and compatibility with modern accelerators (NVIDIA, 2026). Given a block size $B$, the sequence is partitioned into $m = L/B$ contiguous blocks along the token dimension. A block-level mask $\mathbf{M}_b \in \{0, 1\}^{m \times m}$ specifies which block pairs are allowed to interact; lifting $\mathbf{M}_b$ to the token level yields $\widetilde{\mathbf{M}} \in \{0, 1\}^{L \times L}$ in Eq. (3) by setting all entries in a block $(p, q)$ to zero whenever $\mathbf{M}_b[p, q] = 0$.

To quantify how much of the dense attention distribution is preserved under a given mask, we use the recall metric (Jiang et al., 2024a; Zhuang et al., 2025)

$$\mathrm{Recall}(\mathbf{W}, \mathbf{M}) = \frac{\sum_{i,j} \mathbf{W}[i,j]\mathbf{M}[i,j]}{\sum_{i,j} \mathbf{W}[i,j]}. \quad (4)$$

A higher recall indicates that the sparse pattern retains a larger portion of the attention mass (Jiang et al., 2024a).

## 3. Motivating Observations

**Observation 1**: *Head sparsifiability is highly heterogeneous, and non-sparse heads govern the speed–quality trade-off.*

To characterize the sparsifiability landscape, we rank heads by the top-$k$ required to reach a fixed recall threshold (0.8)

and visualize their distribution. As shown in Figure 3, across multiple models, a large fraction of heads are *non-sparse*, requiring substantially larger top-$k$ to attain high recall. On average, the top-$k$ needed to reach recall $= 0.8$ is around 26% and 44% for the models in Figure 3. These non-sparse heads impose a high compute floor and severely limit attainable speedups. To further illustrate this heterogeneity, we take one layer of Wan2.1-1.3B (Wan-AI, 2025) as a case study and plot head-wise recall curves over top-$k$. As shown in Figure 4, heads split into two regimes: (i) *sparse heads* whose recall rapidly saturates at small top-$k$, indicating concentrated attention mass; and (ii) *non-sparse heads* whose recall grows nearly linearly with top-$k$, indicating dispersed attention mass where aggressive sparsity steadily degrades retention. Finally, under a fixed top-$k$=0.05, restoring *only* the non-sparse heads (recall $< 0.8$) to full attention yields substantially larger gains in VBench (Huang et al., 2024) than restoring sparse heads (Figure 5).

**Observation 2**: *Frame-block similarity is pervasive along block-rows and block-columns, and this similarity is well captured by lightweight linear calibration.*

We identify two prevalent types of *frame-block similarity* in attention weights: block-row similarity and block-column similarity. In Figure 6, we highlight both patterns on two representative non-sparse heads, where blocks within the same block-row/column exhibit highly similar attention-weight distributions, manifested as high *in-block row-wise Pearson* correlations (Concretely, for two frame-blocks $\mathbf{X}, \mathbf{Y} \in \mathbb{R}^{\ell \times \ell}$, we compute $\rho(\mathbf{X}, \mathbf{Y}) = \frac{1}{\ell} \sum_{s=0}^{\ell-1} \mathrm{corr}(\mathbf{X}[s, :], \mathbf{Y}[s, :])$, i.e., Pearson correlation on each row vector and then average over rows) (Pearson, 1895). To verify that this phenomenon is widespread, we profile heads (including the denoising step dimension) from Wan2.1-1.3B and fit a simple *in-block row-wise* least-squares scaling calibration: using a random frame-block $\mathbf{B}_{i_0, j_0}$ in each block-row/column as a *prototype*, we map it to another block $\mathbf{W}_{i,j}$ by solving, for each row $s$, $\min_{a_s} \|\mathbf{W}_{i,j}[s, :] - (a_s \mathbf{B}_{i_0, j_0}[s, :])\|_2^2$ and forming $\widehat{\mathbf{W}_{i,j}}[s, :] = a_s \mathbf{B}_{i_0, j_0}[s, :]$. We then report, for each head, the mean $\rho(\mathbf{W}_{i,j}, \widehat{\mathbf{W}_{i,j}})$ (denoted $|R|$) and the relative MSE for attention weight reconstruction: $\mathrm{rMSE} = \frac{\|\mathbf{W} - \widehat{\mathbf{W}}\|_F^2}{\|\mathbf{W}\|_F^2}$ ($\widehat{\mathbf{W}}$ is formed by concatenating $\widehat{\mathbf{W}}_{i,j}$ over all $i, j$). The results in Figure 6 (c) show that this low-cost diagonal (row-wise) calibration explains most inter-block variation, supporting a low-complexity reuse structure.

It is worth noting that this similarity is ultimately rooted in temporal redundancy of video content: nearby frames induce correlated query-key representations and hence similar frame-block attention distributions. Our contribution is not to rediscover temporal redundancy, but to characterize its concrete block-structured manifestation in video DiT

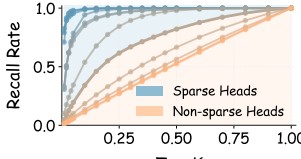

*Figure 4.* Head-wise sparsity variation of Wan2.1-1.3B at step 0 in layer 0.

| model | VBench |
| --- | --- |
| Full Attention | 83.2 |
| Res. sparse | 81.4 |
| Res. non-sparse | 83.0 |

*Figure 5.* Impact of sparse vs. non-sparse heads on video quality. "Res." denotes restoring the corresponding heads to full attention.

*Table 2.* Latent video length setup used in our experiments.

| Model | $n$ | $\ell$ | $L = n\ell$ |
| --- | --- | --- | --- |
| Wan2.1-1.3B | 21 | 1560 | 32760 |
| CogVideoX1.5-5B | 11 | 4080 | 44880 |

attention and turn it into efficient prototype reuse.

**Observation 3**: *The best approximation varies across denoising timesteps, layers, and heads.*

As shown in Figure 7, under matched compute budgets we compare three approximations—sparse attention (*token-pair redundancy*) and prototype reuse with row/column *frame-block similarity*—by measuring how well they reproduce the original attention output $\mathbf{O}$. Sorting heads by their top-$k$ recall, we observe a clear transition: sparse attention works well for highly sparsifiable heads, while prototype reuse becomes increasingly favorable as sparsifiability decreases. To make this separation explicit, Figure 8 visualizes the preferred strategy across heads and denoising timesteps, showing that the optimal choice shifts systematically with not only layer and head, but also denoising steps.

These observations suggest two key takeaways.

***Takeaway 1 (Obs. 1 & 2).*** Beyond *token-pair redundancy*, video DiTs exhibit pervasive *frame-block similarity*, which can be exploited via *prototype reuse* with *linear calibration*.

***Takeaway 2 (Obs. 2 & 3).*** A unified approximation is suboptimal: the best strategy depends on timestep, layer, and head, calling for fine-grained selection.

## 4. Method

We propose **EchoAttention**, an acceleration framework that utilizes the structured frame-block similarity in video DiTs. Figure 9 shows the overview of our method. Firstly, we introduce the structure of EchoAttention, which includes *Sparse* and *Echo* operators, respectively used to efficiently characterize sparse heads and non-sparse heads (Figure 9 (a)). Then, we construct a constrained three-stage distillation framework that allows EchoAttention to learn the most suitable operator stably and accurately at the *timestep-layer-*

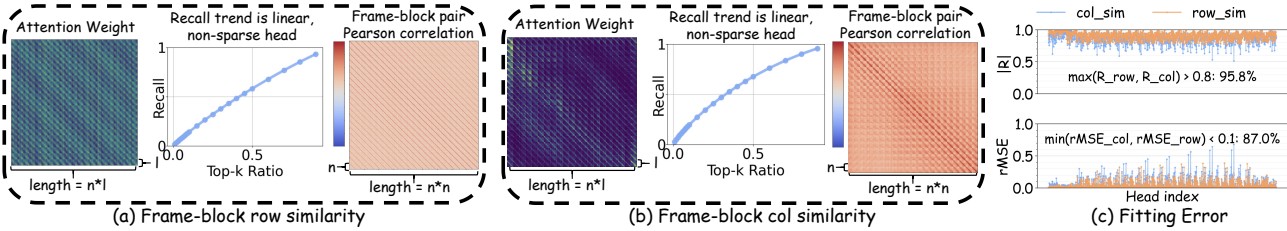

*Figure 6. Frame-block similarity* visualization on Wan2.1-1.3B's heads. We present more examples and results on the MM-DiT model in Appendix E. (a) and (b): the left shows the attention weights of this row/column similar head, showing a coarse frame-block similar pattern; the middle illustrates the recall growing linearly with top-k (low *token-pair redundancy*), indicating it is a non-sparse head; the right shows row-wise Pearson correlation between any *frame-block pair*. Higher values indicate that a per-row linear calibration fits well. The correlation matrices compute pairwise Pearson over the $n \times n$ blocks (row-major flattening): high (near-)diagonal bands suggest within-row block similarity, while strong block-diagonal structure suggests within-column similarity. (c) shows the mean $|R|$ and rMSE across Wan2.1-1.3B heads: mean $|R|$ is near 1 for most heads, and rMSE is often below 0.1, indicating strong linear reconstructability.

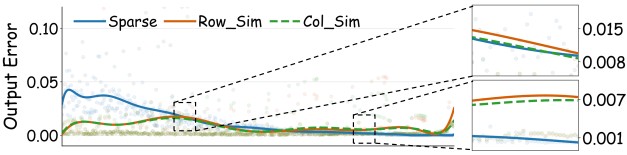

*Figure 7.* rMSE distribution between $\widehat{\mathbf{O}} = \widehat{\mathbf{W}}\mathbf{V}$ and the full-attention output $\mathbf{O}$, where $\widehat{\mathbf{W}}$ is approximated by sparse attention (top-$k$=0.05), frame-block row, or frame-block column similarity. Heads are sorted by the sparse-attention recall (ascending).

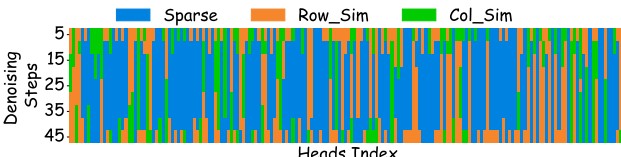

*Figure 8.* Head-wise preference over similarity types, varying across layers/heads and denoising timesteps.

*head* granularity (Figure 9 (b)).

### 4.1. EchoAttention Architecture

As shown on the left of Figure 9 (a), EchoAttention consists of a trainable two-level gating router and three operators: one *Sparse* operator and two *Echo* operators (*Echo-Row* and *Echo-Col*). The first-level gate $g_{\text{sim}}$ decides whether a head uses *Sparse* or *Echo* OP; the second gate $g_{\text{row}}$ selects *Echo-Row* or *Echo-Col* within *Echo OP*. For a head, denote the branch outputs by $\mathbf{O}^{\text{sp}}, \mathbf{O}^{\text{row}}, \mathbf{O}^{\text{col}} \in \mathbb{R}^{L \times d}$. The output of the head is a two-level gated output:

$$\mathbf{O} = (1 - g_{\text{sim}})\,\mathbf{O}^{\text{sp}} + g_{\text{sim}}\Big(g_{\text{row}}\mathbf{O}^{\text{row}} + (1 - g_{\text{row}})\mathbf{O}^{\text{col}}\Big). \quad (5)$$

#### 4.1.1. SPARSE OPERATOR.

We use a standard block-sparse pattern $\widetilde{\mathbf{M}}$ and compute

$$\mathbf{O}^{\text{sp}} = \text{softmax}\left(\frac{\mathbf{Q}\mathbf{K}^{\top}}{\sqrt{d}} + \log \widetilde{\mathbf{M}}\right)\mathbf{V}. \quad (6)$$

The mask $\widetilde{\mathbf{M}}$ is determined by lightweight block mean pooling and top-$k$ selection, as detailed in Appendix A.2.

#### 4.1.2. ECHO OPERATOR.

For the *Echo OP*, we order video tokens frame-wise with $L = n\ell$ and the full-attention output for query frame $i$ is

$$\mathbf{O}_i = \sum_{j=0}^{n-1} \mathbf{W}_{i,j}\mathbf{V}_j, \qquad \mathbf{W}_{i,j} \in \mathbb{R}^{\ell \times \ell}. \quad (7)$$

As suggested in Obs. 2, we approximate each attention weight frame-block $\mathbf{W}_{i,j} \in \mathbb{R}^{\ell \times \ell}$ by a reusable *prototype* $\mathbf{B}$ and a lightweight diagonal calibration $\mathbf{E}$:

For *Echo-Col*, we use $\mathbf{B}_j$ in each block-column and a *left-diagonal* form (query-side) in implementation:

$$\widehat{\mathbf{W}}_{i,j}^{\text{col}} = \mathbf{E}_{i,j}\mathbf{B}_j, \qquad \mathbf{E}_{i,j} = \text{diag}(\mathbf{e}_{i,j}), \quad (8)$$

$$\mathbf{O}_i^{\text{col}} \triangleq \sum_{j=0}^{n-1} \widehat{\mathbf{W}}_{i,j}^{\text{col}}\mathbf{V}_j = \sum_{j=0}^{n-1} \mathbf{E}_{i,j}\big(\mathbf{B}_j\mathbf{V}_j\big), \quad (9)$$

while for *Echo-Row* we use $\mathbf{B}_i$ in each block-row, with a *right-diagonal* (key-side) calibration:

$$\widehat{\mathbf{W}}_{i,j}^{\text{row}} = \mathbf{B}_i\mathbf{E}_{i,j}, \qquad \mathbf{E}_{i,j} = \text{diag}(\mathbf{e}_{i,j}), \quad (10)$$

Equivalently, $(\mathbf{B}_i\mathbf{E}_{i,j})\mathbf{V}_j = \mathbf{B}_i(\mathbf{E}_{i,j}\mathbf{V}_j)$, where $\mathbf{E}_{i,j}$ performs diagonal gating on $\mathbf{V}_j$ for cross-frame value aggregation, and $\mathbf{B}_i$ is applied once to the aggregated values.

$$\mathbf{O}_i^{\text{row}} \triangleq \sum_{j=0}^{n-1} \widehat{\mathbf{W}}_{i,j}^{\text{row}}\mathbf{V}_j = \mathbf{B}_i\Big(\sum_{j=0}^{n-1} \mathbf{E}_{i,j}\mathbf{V}_j\Big), \quad (11)$$

**Algorithm Process.** Let $\mathbf{Q}_i, \mathbf{K}_j, \mathbf{V}_j \in \mathbb{R}^{\ell \times d}$ denote the frame-wise tensors. As shown in Figure 9 (a), *Echo-Row* instantiates Eq. (11) in three steps;

*Step 1: Compute prototype* $\mathbf{B}_i$. From Obs. 2, blocks within a block-row are highly similar; we thus pick a fixed reference block as the prototype:

$$\mathbf{B}_i = \text{softmax}\big(\mathbf{Q}_i\mathbf{K}_0^{\top}/\sqrt{d}\big) \in \mathbb{R}^{\ell \times \ell}. \quad (12)$$

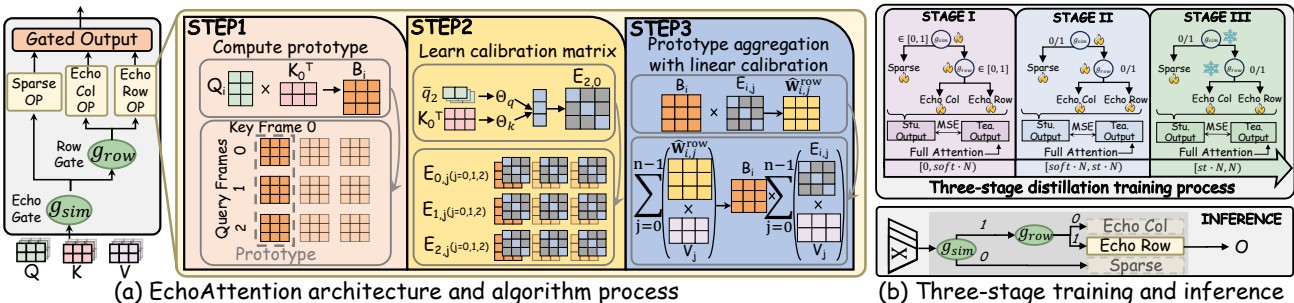

*Figure 9.* The overview of EchoAttention.

Empirically, selecting the first key frame ($j = 0$) as the prototype is simple and effective, see §5.5 for sensitivity.

*Step 2: Learn the calibration diagonal matrix $\mathbf{E}_{i,j}$.* We parameterize the diagonal calibration by token-wise weights $\mathbf{e}_{i,j} \in \mathbb{R}^\ell$. Let $\bar{\mathbf{q}}_i \triangleq \frac{1}{\ell} \sum_{s=0}^{\ell-1} \mathbf{Q}_i[s] \in \mathbb{R}^d$ be a frame summary and introduce two learnable low-dimensional projection modules with parameters $\mathbf{\Theta}_q, \mathbf{\Theta}_k \in \mathbb{R}^{d \times d_{proj}}$ ($d_{proj}$=16). We compute $\mathbf{E}_{i,j}$ by[1]

$$\mathbf{u}_i = \mathbf{\Theta}_q \bar{\mathbf{q}}_i \in \mathbb{R}^{d_{proj}}, \qquad \mathbf{Z}_j = \mathbf{K}_j \mathbf{\Theta}_k \in \mathbb{R}^{\ell \times d_{proj}}, \quad (13)$$

$$\boldsymbol{e}_{i,j} = \exp\left(\frac{1}{\sqrt{d_{proj}}} \mathbf{Z}_j \mathbf{u}_i\right) \in \mathbb{R}_+^\ell, \qquad \mathbf{E}_{i,j} = \mathrm{diag}(\boldsymbol{e}_{i,j}). \quad (14)$$

$\mathbf{E}_{i,j}$ jointly captures inter-block diagonal calibration and compensates the normalization mismatch introduced by using a single prototype. We then enforce a global row normalization over the concatenated frame-block row:

$$\mathbf{c}_i \triangleq \sum_{j=0}^{n-1} \mathbf{e}_{i,j} \in \mathbb{R}_+^\ell, \qquad \mathbf{r}_i \triangleq \mathbf{B}_i \mathbf{c}_i + \epsilon \in \mathbb{R}_+^\ell, \quad (15)$$

$$\mathbf{R}_i \triangleq \mathrm{diag}(\mathbf{r}_i), \qquad \widetilde{\mathbf{B}}_i \triangleq \mathbf{R}_i^{-1} \mathbf{B}_i. \quad (16)$$

where $\epsilon = 10^{-4}$ maintains the numerical stability.

*Step 3: Prototype aggregation with linear calibration.* To instantiate Eq. (11), we use the diagonal matrix $\{\mathbf{E}_{i,j}\}$ as the linear calibration to aggregate value $\mathbf{V}$ across frames:

$$\mathbf{S}_i \triangleq \sum_{j=0}^{n-1} \mathbf{E}_{i,j} \mathbf{V}_j \in \mathbb{R}^{\ell \times d}, \quad \mathbf{S}_i[s] = \sum_{j=0}^{n-1} \mathbf{e}_{i,j}[s] \mathbf{V}_j[s], \quad (17)$$

Finally, we apply the reused prototype $\widetilde{\mathbf{B}}_i$ once to obtain:

$$\mathbf{O}_i^{\mathrm{row}} = \widetilde{\mathbf{B}}_i \mathbf{S}_i = \widetilde{\mathbf{B}}_i \left( \sum_{j=0}^{n-1} \mathbf{E}_{i,j} \mathbf{V}_j \right), \quad (18)$$

*Echo-Col* follows an analogous three-step derivation with left-diagonal mixing and is deferred to Appendix A.1. EchoAttention can also cover *joint 3D attention* with text modality in MM-DiT, as also detailed in Appendix A.3.

---

[1]By default, we use numerically stable safe-exp (max-subtraction) to avoid overflow.

**Complexity.** Step 1 computes $n$ prototypes $\mathbf{B}_i$ [2], costing $O(n\ell^2 d)$. Step 2 forms logits for all $(i, j, s)$ via a $d_{proj}$ dim inner product, costing $O(n^2 \ell d_{proj})$, and the elementwise gating plus normalization $(\mathbf{c}_i, \mathbf{r}_i, \widetilde{\mathbf{B}}_i)$ adds $O(n^2 \ell + n\ell^2)$. Step 3 aggregates $\mathbf{S}_i = \sum_j \mathbf{E}_{i,j} \mathbf{V}_j$ with *diagonal* $\mathbf{E}_{i,j}$ and applies $\widetilde{\mathbf{B}}_i \mathbf{S}_i$, costing $O(n^2 \ell d) + O(n\ell^2 d)$. Since $n \ll \ell$ in long-video generation (e.g., as in Table 2, $n \sim 10^0 - 10^1$ and $\ell \sim 10^3 - 10^4$), the $O(n\ell^2 d)$ terms dominate the $O(n^2 \ell d)$, yielding an overall complexity of $O(n\ell^2 d)$.

### 4.2. Three-Stage Distillation with Routing

Figure 9 (b) illustrates the training and inference pipeline of EchoAttention. Training routed operators can collapse without proper scheduling, so we distill from the full-attention teacher with a two-level gate and a three-stage route-annealing schedule.

Each attention layer $m$ maintains its own gate tables, denoted by $g_{\mathrm{sim}}^{(m)}, g_{\mathrm{row}}^{(m)} \in \mathbb{R}^{T \times H}$. For clarity, we omit the layer index. $g_{\mathrm{sim}}, g_{\mathrm{row}}$ are continuous in training Stage I and become discretized in Stage II–III and inference.

**Stage I (Soft Mix)** evaluates Eq. (5) with continuous gates $g_{\mathrm{sim}}, g_{\mathrm{row}} \in [0, 1]$. It ensures all three branches receive gradients, especially to avoid the routing prematurely favoring one branch in the early stages of training, which may prevent the other branches from learning effective approximations.

**Stage II (Straight-Through (Bengio et al., 2013))** discretizes gates by thresholds

$$g_{\mathrm{sim}}^{\mathrm{hard}} = \mathbb{I}[g_{\mathrm{sim}} > \tau_{\mathrm{sim}}], \quad g_{\mathrm{row}}^{\mathrm{hard}} = \mathbb{I}[g_{\mathrm{row}} > \tau_{\mathrm{row}}], \quad (19)$$

and evaluates Eq. (5) using $(g_{\mathrm{sim}}^{\mathrm{hard}}, g_{\mathrm{row}}^{\mathrm{hard}})$, while backpropagating through the underlying soft gates (straight-through estimator). This stage promotes the gradual discretization of routing and enhances branch specialization.

**Stage III (Hard Route)** freezes parameters $(g_{\mathrm{sim}}^{\mathrm{hard}}, g_{\mathrm{row}}^{\mathrm{hard}})$ into a deterministic lookup table and only trains other learnable parameters, which further strengthens branch training

---

[2]$\mathbf{B}_i$ and $\mathbf{B}_i \mathbf{S}_i$ are computed in a FlashAttention-style streaming manner, as shown in Appendix A.2.

under stable routing.

Let $N$ be the total number of distillation steps. We use two scalars $soft \in (0,1)$ and $st \in (0,1)$ to specify the stage boundaries: Stage I uses steps $[0, \ soft \cdot N)$, Stage II uses steps $[soft \cdot N, \ st \cdot N)$, and Stage III uses steps $[st \cdot N, \ N)$.

**Distillation Loss.** We distill the student by matching the teacher's noise prediction with a standard MSE loss:

$$\mathcal{L}_{\text{distill}} = \mathbb{E}_{(\mathbf{x}_t, t)} \left[ \left\| \hat{\epsilon}_S(\mathbf{x}_t, t) - \text{nograd}(\hat{\epsilon}_T(\mathbf{x}_t, t)) \right\|_2^2 \right]. \quad (20)$$

### 4.3. Implementation

**Kernels.** To realize end-to-end acceleration, we implement custom Triton (Tillet et al., 2019) kernels for EchoAttention, covering both forward and backward passes. The kernels fuse the key operations in Sparse and Echo to maximize the computing capacity and balance workloads. Meanwhile, we implement *Echo OP* in *FlashAttention* (Dao; Shah et al., 2024; Dao et al., 2022) manner to improve memory locality. We provide kernel and class details in Appendix A.2.

**Training Setup.** We distill for 3000 steps with a batch size of 64. We train the gating parameters and the lightweight embedding modules $\boldsymbol{\Theta}_q, \boldsymbol{\Theta}_k$ used by Echo, and additionally fine-tune all linear layers via LoRA (Hu et al., 2022) with rank $r = 128$. For more training details, see Appendix C.

## 5. Experiments

### 5.1. Experimental Setup

**Models and Datasets.** We evaluate EchoAttention on two representative video diffusion models that use 3D full attention: *Wan2.1-1.3B* (Wan-AI, 2025), and *CogVideoX1.5-5B* (zai-org, 2024) (MM-DiT). For inference, *Wan2.1-1.3B* generates 5-second 480p videos, and *CogVideoX1.5-5B* generates 5-second 720p videos. All models use 50 denoising steps. We also evaluate longer generation and other backbones to assess scalability and compatibility (in Appendix D). We distill from the full-attention teacher using a dataset curated from public sources, including OpenSoraPlan (LanguageBind, 2024), MiraData (TencentARC, 2024), and Vript (Mutonix, 2024), covering diverse content.

**Baselines.** We compare our method with state-of-the-art (SOTA) sparse attention methods: *VSA* (Zhang et al., 2025c) and *SLA* (Zhang et al., 2025a) as static methods, and *SpargeAttn* (Zhang et al., 2025b) as a dynamic method. For *SpargeAttn*, we implement a trainable variant and report two settings: *SpargeAttn-Sparse* and *SpargeAttn-Dense*, thus covering different points on the speed–quality trade-off.

We also design four ablation baselines to validate our techniques: (i) *sparse-only*: all heads use the *Sparse OP*; (ii) *echo-only*: all heads use *Echo OP* (*Echo-Row/Echo-Col*);

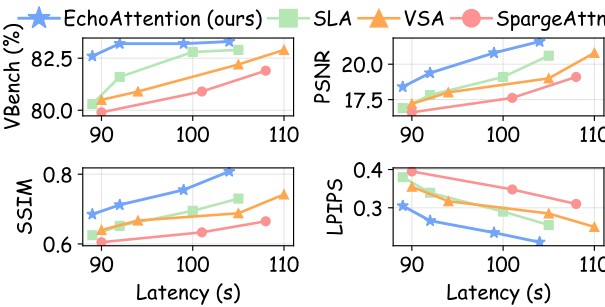

*Figure 10.* The trade-off of quality and latency on Wan2.1-1.3B.

(iii) *echo-full*: heads routed to *Echo OP* are replaced by full attention (to isolate the effect of the *Echo OP*); (iv) *manual-routing*: a handcrafted step–layer–head routing table derived from offline profiling (Obs. 3), without learned gates, to assess the value of automatic routing. The implementation details of all the baselines are in Appendix B.

**Metrics.** Following previous works (Zhang et al., 2025a; Xi et al.; Yang et al., 2025b), we report PSNR (Wang & Bovik, 2009), SSIM (Wang et al., 2004), and LPIPS (Zhang et al., 2018) against full attention as similarity metrics and use *VBench* (Huang et al., 2024) as an aggregate quality metric (Peng et al., 2024; Min et al., 2024). For efficiency, we report end-to-end inference *latency* and *speedup*, measured on RTX 5090 GPUs (NVIDIA, 2025).

**Hyperparameters.** For the *Sparse* operator, we use block_size=64, use top-$k = 0.05$ for Wan2.1-1.3B, and top-$k = 0.1$ for CogVideoX1.5-5B. For routing, we set $\tau_{\text{sim}} = 0.5$ and $\tau_{\text{row}} = 0.5$ to discretize *Sparse OP* vs. *Echo OP* and *Echo-Row* vs. *Echo-Col*, respectively. For the three-stage schedule, we use $soft = 0.05$ and $st = 0.8$.

### 5.2. Main results

**End-to-End Comparison.** As shown in Table 3, EchoAttention consistently improves over all baselines in the overall speed–quality trade-off. EchoAttention achieves the largest end-to-end speedups ($1.97\times$, $2.42\times$) while maintaining quality close to full attention. In contrast, sparse-attention baselines exhibit a clear trade-off governed by non-sparse heads: to preserve quality, they must adopt conservative retention (e.g., *SpargeAttn-Dense*), which yields limited speedup, whereas pushing sparsity more aggressively (e.g., *SpargeAttn-Sparse* or high-sparsity settings) leads to noticeable quality degradation. Among static sparse baselines, SLA performs best overall; however, its linear compensation introduces additional overhead, limiting its speedup at similar quality levels.

**Quality–Speed Trade-off.** We evaluate the quality–speed frontier on Wan2.1-1.3B, as shown in Figure 10. EchoAttention consistently dominates the Pareto frontier (Miettinen,

*Table 3.* Quantitative evaluation results of quality and latency for EchoAttention and other methods, where Sparsity = 1 - top-$k$.

| Method | VBench (%) ↑ | PSNR ↑ | SSIM ↑ | LPIPS ↓ | Sparsity | Latency (s) | Speedup |
|---|---|---|---|---|---|---|---|
| Wan2.1-1.3B (Full Attention) | 83.2 | - | - | - | - | 181 | 1.00× |
| + VSA | 82.2 | 18.01 | 0.6671 | 0.3171 | 89.8% | 105 | 1.72× |
| + SLA | 82.8 | 17.82 | 0.6517 | 0.3392 | 90.0% | 100 | 1.81× |
| + SpargeAttn-Sparse | 81.9 | 17.62 | 0.6333 | 0.3479 | 90.8% | 101 | 1.79× |
| + SpargeAttn-Dense | 82.7 | 19.05 | 0.6912 | 0.2925 | 79.9% | 116 | 1.56× |
| + EchoAttention (**ours**) | **83.2** | **19.38** | **0.7121** | **0.2661** | **95.0%** | **92** | **1.97×** |
| CogVideoX1.5-5B (Full Attention) | 81.9 | - | - | - | - | 770 | 1.00× |
| + VSA | 81.2 | 17.88 | 0.6592 | 0.3210 | 84.9% | 357 | 2.16× |
| + SLA | 81.5 | 18.17 | 0.6723 | 0.3035 | 85.0% | 352 | 2.19× |
| + SpargeAttn-Sparse | 81.0 | 17.22 | 0.6239 | 0.3519 | 85.8% | 351 | 2.19× |
| + SpargeAttn-Dense | 81.6 | 19.95 | 0.7366 | 0.2317 | 71.5% | 418 | 1.84× |
| + EchoAttention (**ours**) | **82.0** | **20.19** | **0.7431** | **0.2391** | **90.0%** | **318** | **2.42×** |

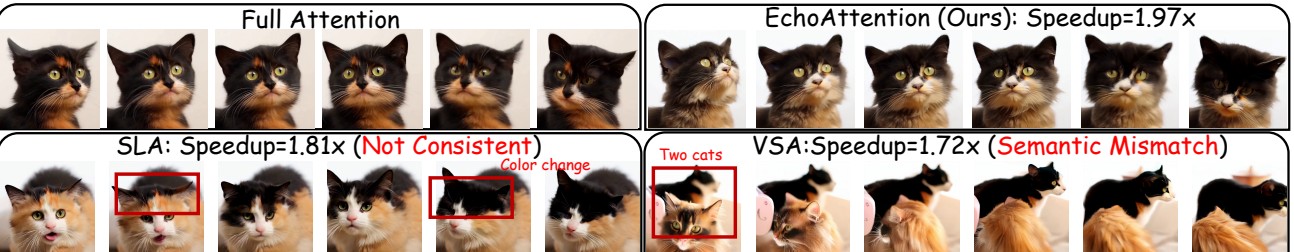

*Figure 11.* Prompt 'A cat' on Wan2.1-1.3B. More examples are in Appendix F.

*Table 4.* Ablation studies of EchoAttention on Wan2.1-1.3B.

| Method | VB↑ | PSNR↑ | SSIM↑ | LPIPS↓ | Lat. | Spd. |
|---|---|---|---|---|---|---|
| Full Attention | 83.2 | - | - | - | 181 | 1.0× |
| *sparse-only* | 81.5 | 17.02 | 0.6173 | 0.3634 | 92 | 1.97× |
| *echo-only* | 81.3 | 16.72 | 0.5998 | 0.3873 | 93 | 1.94× |
| *echo-full* | 83.0 | 19.61 | 0.7119 | 0.2586 | 121 | 1.49× |
| *manual-rt.* | 82.9 | 18.33 | 0.6791 | 0.2983 | 93 | 1.94× |
| **Ours** | **83.2** | 19.38 | 0.7121 | 0.2661 | **92** | **1.97×** |

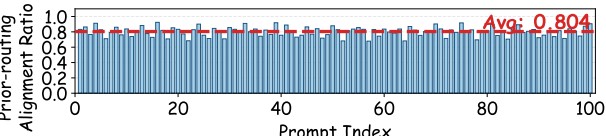

*Figure 12.* Alignment ratio between the learned routing and the prior-implied routing.

1999). In the high-speedup regime, pure sparsification degrades rapidly, whereas EchoAttention degrades more gracefully and maintains acceptable quality at larger speedups. In the low-speedup regime, EchoAttention routes heads to the relatively lower-compute *Echo OP* and achieves higher speed under nearly identical quality. Overall, the results show that purely sparse-based approaches face a speedup ceiling when maintaining high fidelity, while EchoAttention breaks this bottleneck by additionally exploiting frame-block similarity on non-sparse, high-impact heads.

### 5.3. Ablation Studies

We ablate both the operator design and the routing mechanism. The results are shown in Table 4.

**Operator Ablations.** Neither *sparse-only* nor *echo-only* matches our full method, supporting Obs. 3: different heads favor different operators, and a unified strategy is suboptimal. *Echo-full* achieves a quality metric close to our method

but is noticeably slower, indicating that the *Echo* operator recovers most of the needed information while being substantially more efficient than full attention.

**Routing Ablations.** *Manual-routing*, based on offline profiling, provides moderate gains but is consistently worse than learned routing. This suggests that simple error-based heuristics do not fully capture operator suitability, highlighting the importance of our learnable routing.

### 5.4. Case Study

**Routing Statistics.** We summarize the learned routing and compare head recall under a matched top-$k = 0.05$. About 36% of heads are routed to the *Echo OP* (21% *Echo-Row*, 15% *Echo-Col*). In Figure 12, we further sample 100 evaluation prompts and measure the alignment between the operator choice implied by the Obs. 3 prior and our learned routing. Across prompts, the average alignment ratio exceeds 80%, supporting that operator preferences are intrinsic and validating the effectiveness of our learned routing.

_Table 5._ Sensitivity test for training stages.

| Schedule Strategy | VBench ↑ | PSNR ↑ | SSIM ↑ | LPIPS ↓ |
|---|---|---|---|---|
| Always soft (soft=1, st=1) | 82.5 | 18.03 | 0.6711 | 0.3091 |
| Always st (soft=0, st=1) | 82.6 | 19.21 | 0.7022 | 0.2754 |
| No hard (soft=0.05, st=1) | 82.8 | 19.06 | 0.6997 | 0.2865 |
| soft=0.1, st=0.8 | 83.1 | 19.24 | 0.7086 | 0.2718 |
| soft=0.1, st=0.7 | 83.2 | 19.11 | 0.7069 | 0.2742 |
| soft=0.05, st=0.7 | 83.1 | **19.56** | 0.7108 | **0.2623** |
| soft=0.2, st=0.8 | 83.1 | 18.94 | 0.7015 | 0.2796 |
| **soft=0.05, st=0.8 (Ours)** | **83.2** | 19.38 | **0.7121** | 0.2661 |

_Table 6._ Sensitivity test for prototype choice methods.

| Prototype Strategy | VBench ↑ | PSNR ↑ | SSIM ↑ | LPIPS ↓ |
|---|---|---|---|---|
| Random | 82.9 | 18.61 | 0.6924 | 0.2847 |
| Diagonal | 83.2 | 19.13 | 0.7056 | 0.2748 |
| Pooling | 82.8 | 19.21 | 0.7101 | 0.2702 |
| Top-1 Similarity | 83.0 | 18.85 | 0.7013 | 0.2782 |
| **Fixed 0-index (Ours)** | **83.2** | **19.38** | **0.7121** | **0.2661** |

**Visual Examples.** Figure 11 shows a generation example from Wan2.1-1.3B. EchoAttention produces higher visual fidelity, stronger consistency and better semantic adherence than the baselines. Additional examples are provided in Appendix F.

### 5.5. Sensitivity Analysis

**Training Schedule.** In Table 5, we compare our three-stage distillation with three extremes: 1) Always-soft leads to noticeable quality drops due to the train–inference gap, while 2) always-ST tends to collapse routing (95% of heads collapse to the _Sparse OP_), leading to degraded quality. 3) No hard is more stable than always-soft, but it still underperforms due to the lack of a fixed hard routing table that provides a stable training phase. Overall, the three-stage schedule yields the best deployment performance. Meanwhile, we also investigated the sensitivity of the three-stage approach to hyperparameters _soft_ and _st_. The results show that, within a reasonable range, performance remains relatively stable, demonstrating that our method is practically insensitive to these hyperparameters.

**Prototype Choice.** We set random, diagonal, mean pooling, and _Top-1 Similarity_ prototype selections and conducted the comparative experiments in Table 6. _Top-1 Similarity_ selects the prototype adaptively by a lightweight similarity score and uses the most similar frame as the prototype in Step 1 (Appendix B). The results show that the fixed 0-index selection we used is among the best and is sufficient.

## 6. Conclusion

We presented **EchoAttention**, an efficient attention framework for video diffusion Transformers that goes beyond _token-pair sparsity_ by exploiting _frame-block similarity_. EchoAttention combines a block-sparse operator with an _Echo_ operator and learns fine-grained routing at the _timestep–layer–head_ level via a three-stage distillation schedule. This design reduces the dominant attention cost while preserving fidelity, consistently improving the speed–quality frontier across multiple public video DiTs.

## Acknowledgments

This work is supported by National Natural Science Foundation of China (U23B2048, 62402011), Fundamental and Interdisciplinary Disciplines Breakthrough Plan of the Ministry of Education of China (JYB2025XDXM108), ByteDance-PKU joint program, and High-performance Computing Platform of Peking University. Fangcheng Fu and Bin Cui are the corresponding authors.

## Impact Statement

This paper presents work whose goal is to advance the field of Machine Learning. There are many potential societal consequences of our work, none of which we feel must be specifically highlighted here.

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

# A. EchoAttention Design Details

## A.1. Derivation and Algorithm Process of *Echo-Col*

*Echo-Col* is a column-wise counterpart of *Echo-Row* (Eq. (9)), tailored to heads exhibiting block-column similarity (Obs. 2). We use the same blockwise decomposition with $L = n\ell$ and frame-wise tensors $\mathbf{Q}_i, \mathbf{K}_j, \mathbf{V}_j \in \mathbb{R}^{\ell \times d}$. *Echo-Col* shares prototypes along each block-column and applies a left-diagonal calibration:

$$\widehat{\mathbf{W}}_{i,j}^{\mathrm{col}} = \mathbf{E}_{i,j}\,\mathbf{B}_j, \qquad \mathbf{E}_{i,j} = \mathrm{diag}(\mathbf{e}_{i,j}), \tag{21}$$

which yields

$$\mathbf{O}_i^{\mathrm{col}} \triangleq \sum_{j=0}^{n-1} \widehat{\mathbf{W}}_{i,j}^{\mathrm{col}} \mathbf{V}_j = \sum_{j=0}^{n-1} \mathbf{E}_{i,j}\big(\mathbf{B}_j \mathbf{V}_j\big). \tag{22}$$

**Step 1: Compute column prototypes $\mathbf{B}_j$.** Under block-column similarity, we can choose a fixed reference query block (e.g., the first query frame) and form all column prototypes in one shot:

$$\mathbf{B}_j \triangleq \mathrm{softmax}\Big(\mathbf{Q}_0 \mathbf{K}_j^\top / \sqrt{d}\Big) \in \mathbb{R}^{\ell \times \ell}, \qquad j = 0, \dots, n-1, \tag{23}$$

because the same frame-row's key block is used, each $\mathbf{B}_j$ is itself an attention block that maintains the original softmax probability distribution.

**Step 2: Learn diagonal calibration $\mathbf{E}_{i,j}$ with query-aligned token weights.** To match the *left-diagonal* form in Eq. (21), the calibration vector $\mathbf{e}_{i,j} \in \mathbb{R}^\ell$ must be indexed by the *query-token* position. We therefore generate $\mathbf{e}_{i,j}[s]$ from *token-wise* query features of frame $i$ and a *frame-summary* key descriptor of frame $j$.

Let $\mathbf{\Theta}_q, \mathbf{\Theta}_k \in \mathbb{R}^{d \times d_{\mathrm{proj}}}$ with $d_{\mathrm{proj}}{=}16$. Define the token-wise projected queries and the frame-level key summary as

$$\mathbf{Z}_i \triangleq \mathbf{Q}_i \mathbf{\Theta}_q \in \mathbb{R}^{\ell \times d_{\mathrm{proj}}}, \qquad \bar{\mathbf{k}}_j \triangleq \frac{1}{\ell} \sum_{s=0}^{\ell-1} \mathbf{K}_j[s] \in \mathbb{R}^d, \qquad \mathbf{u}_j \triangleq \mathbf{\Theta}_k \bar{\mathbf{k}}_j \in \mathbb{R}^{d_{\mathrm{proj}}}. \tag{24}$$

We then parameterize the diagonal weights by

$$\mathbf{e}_{i,j} \triangleq \exp\Big(\frac{1}{\sqrt{d_{\mathrm{proj}}}} \mathbf{Z}_i \mathbf{u}_j\Big) \in \mathbb{R}_+^\ell, \qquad \mathbf{E}_{i,j} \triangleq \mathrm{diag}(\mathbf{e}_{i,j}). \tag{25}$$

As in *Echo-Row*, we use a numerically stable safe-$\exp$ (max-subtraction) to avoid overflow.

For each query frame $i$, we normalize the unnormalized nonnegative weights across all key frames:

$$\mathbf{r}_i \triangleq \sum_{j=0}^{n-1} \mathbf{e}_{i,j} \in \mathbb{R}_+^\ell, \qquad \widetilde{\mathbf{E}}_{i,j} \triangleq \mathrm{diag}(\mathbf{r}_i + \epsilon)^{-1} \mathbf{E}_{i,j}, \tag{26}$$

where $\epsilon = 10^{-4}$ improves numerical stability. By construction, for each query token index $s$ we have $\sum_j \widetilde{\mathbf{E}}_{i,j}[s, s] = 1$.

**Step 3: Prototype aggregation with linear calibration.** We first compute reusable per-column contributions

$$\mathbf{T}_j \triangleq \mathbf{B}_j \mathbf{V}_j \in \mathbb{R}^{\ell \times d}. \tag{27}$$

Then, for each query frame $i$, we mix $\{\mathbf{T}_j\}_{j=0}^{n-1}$ using the normalized left-diagonal weights:

$$\mathbf{O}_i^{\mathrm{col}} = \sum_{j=0}^{n-1} \widetilde{\mathbf{E}}_{i,j} \mathbf{T}_j, \qquad \mathbf{O}_i^{\mathrm{col}}[s] = \sum_{j=0}^{n-1} \frac{\mathbf{e}_{i,j}[s]}{\mathbf{r}_i[s] + \epsilon} \mathbf{T}_j[s]. \tag{28}$$

This exactly matches the left-diagonal form $\widehat{\mathbf{W}}_{i,j}^{\mathrm{col}} = \mathbf{E}_{i,j} \mathbf{B}_j$: $\mathbf{e}_{i,j}[s]$ is indexed by the query-token position $s$ and controls how token $s$ aggregates information across frames.

**Complexity.** Step 1 computes all prototypes $\{\mathbf{B}_j\}_{j=0}^{n-1}$, costing $O(n\ell^2 d)$. Step 2 forms token-wise logits for all $(i, j, s)$ via a $d_{proj}$-dim inner product, costing $O(n^2 \ell d_{proj})$, and the element-wise gating plus normalization $(\mathbf{c}_i, \mathbf{r}_i, \widetilde{\mathbf{E}}_{i,j})$ adds $O(n^2 \ell + n\ell)$. Step 3 computes $\mathbf{T}_j = \mathbf{B}_j \mathbf{V}_j$ and aggregates $\sum_j \widetilde{\mathbf{E}}_{i,j} \mathbf{T}_j$, costing $O(n\ell^2 d) + O(n^2 \ell d)$. Overall, under the typical long-video regime $n \ll \ell$, *Echo-Col* is dominated by $O(n\ell^2 d)$, matching *Echo-Row*.

## A.2. EchoAttention Class and Kernels Design

EchoAttention is implemented as a drop-in multi-head attention module with a two-level router and three operator branches (Sparse / *Echo-Row* / *Echo-Col*). For each transformer layer, the router maintains two per-$(t, h)$ gate tables: $g_{\text{sim}}(t, h)$ selects *Sparse* vs. *Echo*, and $g_{\text{row}}(t, h)$ further selects *Echo-Row* vs. *Echo-Col* inside the Echo branch (Eq. (5)). Echo additionally introduces two learnable low-dimensional projection modules $\mathbf{\Theta}_q, \mathbf{\Theta}_k \in \mathbb{R}^{d \times d_{proj}}$, used to produce the diagonal calibration weights via Eq. (14) (*Echo-Row*) and Eq. (25) (*Echo-Col*). Both *Echo-Row* and *Echo-Col* enforce the global probability preservation described in Eq. (15)–(16) (Row) and Eq. (26) (Col), respectively.

**Class Definition.** Listing 1 sketches the core class structure. We use a fused `echoattn_kernel` that (i) partitions heads by routing, (ii) executes Sparse / *Echo-Row* / *Echo-Col* in a batched manner that maximizes the use of computing capacity, and (iii) merges outputs back, avoiding extra dispatch overhead and large intermediates.

*Listing 1.* Minimal EchoAttention class (core forward, fused kernel).

```
1  class GateTable(nn.Module):
2      def __init__(self, T, num_heads):
3          super().__init__()
4          self.alpha = nn.Embedding(T, num_heads)
5          nn.init.zeros_(self.alpha.weight)
6
7      def soft_gate(self, t_idx):
8          p = torch.sigmoid(self.alpha(torch.tensor([t_idx],
           device=self.alpha.weight.device))[0])
9          return p
10     def st_gate(self, t_idx, th=0.5):
11         p = torch.sigmoid(self.alpha(torch.tensor([t_idx],
           device=self.alpha.weight.device))[0])
12         hard = (p > th).to(p.dtype)
13         return hard.detach() - p.detach() + p
14     def hard_gate(self, t_idx, th=0.5):
15         p = torch.sigmoid(self.alpha(torch.tensor([t_idx],
           device=self.alpha.weight.device))[0])
16         return (p > th).to(p.dtype)
17
18 class EchoAttention(nn.Module):
19     def __init__(self, d, num_heads, T, blkq=64, blkk=64, default_topk=0.05, proj=16):
20         super().__init__()
21         self.d = d
22         self.H = num_heads
23         self.BLKQ, self.BLKK = blkq, blkk
24         self.default_topk = default_topk
25
26         self.g_sim = GateTable(T, num_heads)
27         self.g_row = GateTable(T, num_heads)
28
29         self.Theta_q = nn.Linear(d, proj, bias=False)
30         self.Theta_k = nn.Linear(d, proj, bias=False)
31
32     def forward(self, q, k, v, *, timestep, tokens_per_frame, topk=None, stage):
33         topk = self.default_topk if topk is None else float(topk)
34         if stage == 'soft':
35             sim_gate = self.g_sim.soft_gate(timestep)   # Echo vs Sparse
36             row_gate = self.g_row.soft_gate(timestep)   # Row vs Col inside Echo
37         elif stage == 'st':
38             sim_gate = self.g_sim.st_gate(timestep)     # Echo vs Sparse
39             row_gate = self.g_row.st_gate(timestep)     # Row vs Col inside Echo
40         else:
```

```
41          sim_gate = self.g_sim.hard_gate(timestep)    # Echo vs Sparse
42          row_gate = self.g_row.hard_gate(timestep)    # Row vs Col inside Echo
43      # fused kernel: routes + computes Sparse / Echo-Row / Echo-Col and merges outputs
44      out = echoattn_kernel(
45          q, k, v,
46          sim_gate=sim_gate, row_gate=row_gate,
47          tokens_per_frame=tokens_per_frame,
48          topk=topk,
49          blkq=self.BLKQ, blkk=self.BLKK,
50          theta_q=self.Theta_q.weight,
51          theta_k=self.Theta_k.weight,
52      )
53      return out
```

**Sparse OP Kernel.** We show the pseudocode for the forward and backward pass of *Sparse OP* in Algorithm 1 and 2. *Sparse OP* follows Eq. (6) with a *block* mask $\widetilde{\mathbf{M}}$. Let the block size be $(\mathrm{BLKQ}, \mathrm{BLKK})$ on the query/key axis. We first compute block-level *mean-pooled* queries/keys by averaging tokens within each block:

$$\bar{\mathbf{Q}} = \mathrm{BlockMean}(\mathbf{Q}) \in \mathbb{R}^{B \times H_{\mathrm{sp}} \times m_q \times d}, \quad \bar{\mathbf{K}} = \mathrm{BlockMean}(\mathbf{K}) \in \mathbb{R}^{B \times H_{\mathrm{sp}} \times m_k \times d},$$

where $m_q = L/\mathrm{BLKQ}$ and $m_k = L/\mathrm{BLKK}$. We then form block logits and normalize them into block attention weights

$$\mathbf{S} = \mathrm{softmax}\Big(\bar{\mathbf{Q}}\bar{\mathbf{K}}^{\top}/\sqrt{d}\Big) \in \mathbb{R}^{B \times H_{\mathrm{sp}} \times m_q \times m_k}. \tag{29}$$

For each query block $p \in \{0, \ldots, m_q - 1\}$, we keep the top-$k$ key blocks under $\mathbf{S}[\cdot, \cdot, p, :]$ (with $k = \lceil \gamma m_k \rceil$) and set the binary block mask $\widetilde{\mathbf{M}} \in \{0, 1\}^{m_q \times m_k}$ accordingly. Finally, we apply a fused block-sparse attention kernel using $\widetilde{\mathbf{M}}$:

$$\mathbf{O}^{\mathrm{sp}} = \mathrm{BlockSparseAttn}(\mathbf{Q}, \mathbf{K}, \mathbf{V}; \widetilde{\mathbf{M}}). \tag{30}$$

**Echo-Row OP Kernel.** We show the pseudocode for the forward and backward pass of *Echo-Row OP* in Algorithm 3 and 4. *Echo-Row* instantiates Eq. (18) with three steps. (*Step 1*) compute the row prototype $\mathbf{B}_i = \mathrm{softmax}(\mathbf{Q}_i \mathbf{K}_0^{\top}/\sqrt{d})$ (Eq. (12)) in a *FlashAttention* style; (*Step 2*) predict diagonal weights $\mathbf{e}_{i,j} = \exp(\cdot)$ (Eq. (14)) and apply the global row-normalization $\mathbf{r}_i = \mathbf{B}_i \sum_j \mathbf{e}_{i,j}$, $\widetilde{\mathbf{B}}_i = \mathrm{diag}(\mathbf{r}_i)^{-1}\mathbf{B}_i$ (Eqs. (15)–(16)); (*Step 3*) aggregate $\mathbf{S}_i = \sum_j \mathrm{diag}(\mathbf{e}_{i,j})\mathbf{V}_j$ and output $\mathbf{O}_i^{\mathrm{row}} = \widetilde{\mathbf{B}}_i \mathbf{S}_i$ (Eq. (18)).

**Echo-Col OP Kernel.** We show the pseudocode for the forward and backward pass of *Echo-Col OP* in Algorithm 5 and 6. *Echo-Col* follows Appendix A.1. (*Step 1*) compute column prototypes $\mathbf{B}_j = \mathrm{softmax}(\mathbf{Q}_0 \mathbf{K}_j^{\top}/\sqrt{d})$ (Eq. (23)) in a *FlashAttention* style and the reusable contributions $\mathbf{T}_j = \mathbf{B}_j \mathbf{V}_j$ (Eq. (27)); (*Step 2*) predict $\mathbf{e}_{i,j} = \exp(\cdot)$ and normalize $\widetilde{\mathbf{E}}_{i,j} = \mathrm{diag}(\mathbf{r}_i)^{-1}\mathrm{diag}(\mathbf{e}_{i,j})$ (Eq. (26)); (*Step 3*) mix $\{\mathbf{T}_j\}$ by left-diagonal weights: $\mathbf{O}_i^{\mathrm{col}} = \sum_j \widetilde{\mathbf{E}}_{i,j}\mathbf{T}_j$ (Eq. (28)).

### A.3. Adaptation for MM-DiT Architecture

Many MM-DiT backbones (e.g., CogVideoX) apply *joint attention* over concatenated text and video tokens. Let the text length be $L_{\mathrm{text}}$ and the video length be $n\ell$, so the total sequence length is $L = L_{\mathrm{text}} + n\ell$. For each head, full attention is

$$\mathbf{W} = \mathrm{softmax}\Big(\mathbf{Q}\mathbf{K}^{\top}/\sqrt{d}\Big) \in \mathbb{R}^{L \times L}, \qquad \mathbf{O} = \mathbf{W}\mathbf{V} \in \mathbb{R}^{L \times d}. \tag{31}$$

Under the ordering "text first, video second", $\mathbf{W}$ admits a $2 \times 2$ block structure

$$\mathbf{W} = \begin{bmatrix} \mathbf{W}^{\mathrm{tt}} & \mathbf{W}^{\mathrm{tv}} \\ \mathbf{W}^{\mathrm{vt}} & \mathbf{W}^{\mathrm{vv}} \end{bmatrix}, \tag{32}$$

where $\mathbf{W}^{\mathrm{tt}} \in \mathbb{R}^{L_{\mathrm{text}} \times L_{\mathrm{text}}}, \mathbf{W}^{\mathrm{tv}} \in \mathbb{R}^{L_{\mathrm{text}} \times (n\ell)}, \mathbf{W}^{\mathrm{vt}} \in \mathbb{R}^{(n\ell) \times L_{\mathrm{text}}}$, and $\mathbf{W}^{\mathrm{vv}} \in \mathbb{R}^{(n\ell) \times (n\ell)}$. Moreover, the video–video block $\mathbf{W}^{\mathrm{vv}}$ can be further viewed as an $n \times n$ grid of $\ell \times \ell$ frame-blocks:

$$\mathbf{W}^{\mathrm{vv}} = \begin{bmatrix} \mathbf{W}_{00} & \cdots & \mathbf{W}_{0(n-1)} \\ \vdots & \ddots & \vdots \\ \mathbf{W}_{(n-1)0} & \cdots & \mathbf{W}_{(n-1)(n-1)} \end{bmatrix}, \qquad \mathbf{W}_{ij} \in \mathbb{R}^{\ell \times \ell}. \tag{33}$$

---

**Algorithm 1** `SparseHeadFwd`.

---

**Require:** $\mathbf{Q}, \mathbf{K}, \mathbf{V} \in \mathbb{R}^{L \times d}$; (BLKQ, BLKK); top-$k$ ratio $\gamma$.
**Ensure:** $\mathbf{O} \in \mathbb{R}^{L \times d}$ and selected block indices $\{\mathcal{N}(p)\}_{p=0}^{m_q-1}$.
 1: $m_q \leftarrow L/\text{BLKQ}; \quad m_k \leftarrow L/\text{BLKK}; \quad k \leftarrow \max(1, \lceil \gamma m_k \rceil)$
 2: **for** $p = 0..m_q - 1$ **do**
 3: $\quad \bar{\mathbf{Q}}[p] \leftarrow \text{mean}(\mathbf{Q}_p) \in \mathbb{R}^d$
 4: **end for**
 5: **for** $q = 0..m_k - 1$ **do**
 6: $\quad \bar{\mathbf{K}}[q] \leftarrow \text{mean}(\mathbf{K}_q) \in \mathbb{R}^d$
 7: **end for**
 8: **for** $p = 0..m_q - 1$ **do**
 9: $\quad \mathbf{S}[p, q] \leftarrow \text{softmax}_q\big(\langle \bar{\mathbf{Q}}[p], \bar{\mathbf{K}}[q] \rangle / \sqrt{d}\big)$
10: $\quad \mathcal{N}(p) \leftarrow \text{TopKIdx}(\mathbf{S}[p, :], k)$
11: **end for**
12: $\mathbf{O} \leftarrow \mathbf{0}$
13: **for** $p = 0..m_q - 1$ **do**
14: $\quad \mathbf{Q}_p \leftarrow \mathbf{Q}[p\,\text{BLKQ} : (p+1)\text{BLKQ}]$
15: $\quad \mathbf{m} \leftarrow -\infty \cdot \mathbf{1} \in \mathbb{R}^{\text{BLKQ}}; \quad \boldsymbol{\ell} \leftarrow \mathbf{0} \in \mathbb{R}^{\text{BLKQ}}; \quad \mathbf{A} \leftarrow \mathbf{0} \in \mathbb{R}^{\text{BLKQ} \times d}$
16: $\quad$ **for** each $q \in \mathcal{N}(p)$ **do**
17: $\quad\quad \mathbf{K}_q \leftarrow \mathbf{K}[q\,\text{BLKK} : (q+1)\text{BLKK}]; \quad \mathbf{V}_q \leftarrow \mathbf{V}[q\,\text{BLKK} : (q+1)\text{BLKK}]$
18: $\quad\quad \mathbf{P} \leftarrow \mathbf{Q}_p \mathbf{K}_q^\top / \sqrt{d} \in \mathbb{R}^{\text{BLKQ} \times \text{BLKK}}$
19: $\quad\quad \mathbf{m}' \leftarrow \max(\mathbf{m}, \text{rowmax}(\mathbf{P}))$
20: $\quad\quad \mathbf{G} \leftarrow \exp(\mathbf{P} - \mathbf{m}'[:, None])$
21: $\quad\quad \boldsymbol{\ell} \leftarrow \exp(\mathbf{m} - \mathbf{m}') \odot \boldsymbol{\ell} + \text{rowsum}(\mathbf{G})$
22: $\quad\quad \mathbf{A} \leftarrow \exp(\mathbf{m} - \mathbf{m}')[:, None] \odot \mathbf{A} + \mathbf{G}\mathbf{V}_q$
23: $\quad\quad \mathbf{m} \leftarrow \mathbf{m}'$
24: $\quad$ **end for**
25: $\quad \mathbf{O}[p\,\text{BLKQ} : (p+1)\text{BLKQ}] \leftarrow \mathbf{A} \oslash \boldsymbol{\ell}[:, None]$
26: **end for**
27: **return** $\mathbf{O}$ and $\{\mathcal{N}(p)\}_{p=0}^{m_q-1}$

---

**Key idea: exact text, Echo video, and consistent cross-group normalization.** EchoAttention is designed for video tokens. For MM-DiT, we keep all text-related interactions *exact* and apply EchoAttention only on the video–video sub-block $\mathbf{W}^{\text{vv}}$. Crucially, we retain a *principled two-group normalization in the approximated model* via a group-wise partition view: the text-group partition is computed exactly, while Echo provides the corresponding video-group normalizer for the approximated video weights. This yields a normalized student attention that is self-consistent under our approximation, and any residual mismatch to the full-attention teacher is corrected by distillation.

**Group-wise decomposition of joint attention (video queries).** Consider a *video* query token with query $\mathbf{q} \in \mathbb{R}^d$. Let $\boldsymbol{\alpha}^{\text{text}} \in \mathbb{R}^{L_{\text{text}}}$ and $\boldsymbol{\alpha}^{\text{vid}} \in \mathbb{R}^{n\ell}$ denote its logits to text keys and video keys:

$$\alpha^{\text{text}}[m] = \tfrac{1}{\sqrt{d}} \mathbf{q}^\top \mathbf{k}_m^{\text{text}}, \qquad \alpha^{\text{vid}}[u] = \tfrac{1}{\sqrt{d}} \mathbf{q}^\top \mathbf{k}_u^{\text{vid}}.$$

Define the (exact) group partition functions

$$Z^{\text{text}} \triangleq \sum_{m=0}^{L_{\text{text}}-1} \exp(\alpha^{\text{text}}[m]), \qquad Z^{\text{vid}} \triangleq \sum_{u=0}^{n\ell-1} \exp(\alpha^{\text{vid}}[u]). \tag{34}$$

Then the exact joint-softmax output for this video query can be written as a mixture of two *within-group normalized* outputs:

$$\mathbf{o} = \lambda \, \mathbf{o}^{\text{text}} + (1 - \lambda) \, \mathbf{o}^{\text{vid}}, \qquad \lambda \triangleq \frac{Z^{\text{text}}}{Z^{\text{text}} + Z^{\text{vid}}}, \tag{35}$$

---

**Algorithm 2** `SparseHeadBwd`.

---

**Require:** $\nabla\mathbf{O} \in \mathbb{R}^{L\times d}$; $\mathbf{Q}, \mathbf{K}, \mathbf{V}$; selected blocks $\{\mathcal{N}(p)\}_{p=0}^{m_q-1}$; per-block softmax probabilities $\mathbf{B}_{p,q} = \text{softmax}(\mathbf{Q}_p\mathbf{K}_q^\top/\sqrt{d})$ over $q \in \mathcal{N}(p)$.
**Ensure:** $\nabla\mathbf{Q}, \nabla\mathbf{K}, \nabla\mathbf{V}$.
  1: $\nabla\mathbf{Q} \leftarrow \mathbf{0}; \quad \nabla\mathbf{K} \leftarrow \mathbf{0}; \quad \nabla\mathbf{V} \leftarrow \mathbf{0}$
  2: **for** $p = 0..m_q - 1$ **do**
  3: $\quad \nabla\mathbf{O}_p \leftarrow \nabla\mathbf{O}[p\,\text{BLKQ} : (p+1)\text{BLKQ}]$
  4: $\quad$ **for** each $q \in \mathcal{N}(p)$ **do**
  5: $\quad\quad \mathbf{K}_q, \mathbf{V}_q$ are the corresponding blocks
  6: $\quad\quad \mathbf{B} \leftarrow \mathbf{B}_{p,q}$
  7: $\quad\quad \nabla\mathbf{V}_q \mathrel{+}= \mathbf{B}^\top\nabla\mathbf{O}_p$
  8: $\quad\quad \nabla\mathbf{B} \leftarrow \nabla\mathbf{O}_p\mathbf{V}_q^\top$
  9: $\quad\quad \mathbf{s} \leftarrow \text{rowsum}(\nabla\mathbf{B} \odot \mathbf{B})$
 10: $\quad\quad \nabla\mathbf{P} \leftarrow (\nabla\mathbf{B} - \mathbf{s}[:, None]) \odot \mathbf{B}$
 11: $\quad\quad \nabla\mathbf{Q}_p \mathrel{+}= \nabla\mathbf{P}\mathbf{K}_q/\sqrt{d}$
 12: $\quad\quad \nabla\mathbf{K}_q \mathrel{+}= \nabla\mathbf{P}^\top\mathbf{Q}_p/\sqrt{d}$
 13: $\quad$ **end for**
 14: $\quad$ Scatter-add $\nabla\mathbf{Q}_p$ into $\nabla\mathbf{Q}[p\,\text{BLKQ} : (p+1)\text{BLKQ}]$
 15: **end for**
 16: **return** $(\nabla\mathbf{Q}, \nabla\mathbf{K}, \nabla\mathbf{V})$

---

where

$$\mathbf{o}^{\text{text}} \triangleq \sum_m \frac{\exp(\alpha^{\text{text}}[m])}{Z^{\text{text}}}\mathbf{v}_m^{\text{text}}, \qquad \mathbf{o}^{\text{vid}} \triangleq \sum_u \frac{\exp(\alpha^{\text{vid}}[u])}{Z^{\text{vid}}}\mathbf{v}_u^{\text{vid}}. \tag{36}$$

This identity is exact and simply re-expresses the joint softmax using group partitions.

**Our approximation.** We keep the text group exact by computing $\mathbf{o}^{\text{text}}$ and $\log Z^{\text{text}}$ with standard attention over text keys (Eq. (36)). For the video group, we approximate $\mathbf{o}^{\text{vid}}$ using EchoAttention on video tokens only (*Echo-Row* or *Echo-Col*, identical to the main method). Importantly, EchoAttention also provides a *nonnegative per-query-token pre-normalization mass* that serves as the video-group partition under our approximation, enabling an *exact* mixture weight *within the approximated model*.

**Echo provides $\widehat{Z}^{\text{vid}}$ as an explicit pre-normalizer.** For a video query frame $i$ and within-frame token index $s$, denote the video-group (approximated) attention weights over video keys by $\widehat{\mathbf{W}}^{\text{vv}}$. Both *Echo-Row* and *Echo-Col* construct $\widehat{\mathbf{W}}^{\text{vv}}$ by first forming an *unnormalized* nonnegative weight row and then applying a row-wise normalization:

$$\widehat{\mathbf{W}}^{\text{vv}}[i, s, \cdot] = \frac{\overline{\mathbf{W}}^{\text{vv}}[i, s, \cdot]}{\widehat{Z}_{i,s}^{\text{vid}}}, \qquad \widehat{Z}_{i,s}^{\text{vid}} \triangleq \sum_u \overline{\mathbf{W}}^{\text{vv}}[i, s, u].$$

Concretely:

- ***Echo-Row*:** the unnormalized block-row is $\overline{\mathbf{W}}_{i,j}^{\text{vv}} = \mathbf{B}_i\mathbf{E}_{i,j}$ (Eq. (10)). Its row-sum over all video keys is exactly the vector $\mathbf{r}_i$ already computed in the main method:

$$\widehat{Z}_{i,s}^{\text{vid}} = \mathbf{r}_i[s], \qquad \mathbf{r}_i \triangleq \mathbf{B}_i\Big(\sum_{j=0}^{n-1}\mathbf{e}_{i,j}\Big) \text{ (Eq. (15))}. \tag{37}$$

- ***Echo-Col*:** the unnormalized block-row is $\overline{\mathbf{W}}_{i,j}^{\text{vv}} = \mathbf{E}_{i,j}\mathbf{B}_j$ (Eq. (21)). Its row-sum over all video keys is exactly the vector $\mathbf{r}_i$ in Eq. (26):

$$\widehat{Z}_{i,s}^{\text{vid}} = \mathbf{r}_i[s], \qquad \mathbf{r}_i \triangleq \sum_{j=0}^{n-1}\mathbf{E}_{i,j}\big(\mathbf{B}_j\mathbf{1}\big) \text{ (Eq. (26))}. \tag{38}$$

---

**Algorithm 3** `EchoRowHeadFwd`.

---

**Require:** $\mathbf{Q}, \mathbf{K}, \mathbf{V} \in \mathbb{R}^{L \times d}$; $\ell$ $(n = L/\ell)$; $\boldsymbol{\Theta}_q, \boldsymbol{\Theta}_k$.
**Ensure:** $\mathbf{O} \in \mathbb{R}^{L \times d}$ and intermediates (`stats`, $\mathbf{e}, \mathbf{r}, \mathbf{S}$).
 1: Reshape $\mathbf{Q}, \mathbf{K}, \mathbf{V} \to \{\mathbf{Q}_i, \mathbf{K}_i, \mathbf{V}_i\}_{i=0}^{n-1}$ with $\mathbf{Q}_i \in \mathbb{R}^{\ell \times d}$.
 2: *Step 1 (prototype logits stats, FlashAttention streaming):*
 3: **for** $i = 0..n-1$ **do**
 4:    $\text{stats}_i \leftarrow \texttt{FlashAttnStats}(\mathbf{Q}_i, \mathbf{K}_0)$ {streaming row-wise max/LSE; no materialization of $\mathbf{B}_i$}
 5: **end for**
 6: *Step 2:* Predict diagonal weights and form $\mathbf{c}_i$:
 7:    $\bar{\mathbf{q}}_i \leftarrow \text{mean}_s(\mathbf{Q}_i) \in \mathbb{R}^d$;   $\mathbf{u}_i \leftarrow \boldsymbol{\Theta}_q \bar{\mathbf{q}}_i \in \mathbb{R}^{d_{proj}}$;   $\mathbf{Z}_j \leftarrow \mathbf{K}_j \boldsymbol{\Theta}_k \in \mathbb{R}^{\ell \times d_{proj}}$
 8:    $\mathbf{e}_{i,j} \leftarrow \exp\big(\mathbf{Z}_j \mathbf{u}_i / \sqrt{d_{proj}}\big) \in \mathbb{R}_+^\ell$;   $\mathbf{c}_i \leftarrow \sum_j \mathbf{e}_{i,j} \in \mathbb{R}_+^\ell$
 9: *Step 3:* Streaming apply of prototype and normalization:
10:    $\mathbf{S}_i[s] \leftarrow \sum_j \mathbf{e}_{i,j}[s] \cdot \mathbf{V}_j[s] \in \mathbb{R}^d$;   $\mathbf{r}_i \leftarrow \texttt{FlashAttnApply}(\mathbf{Q}_i, \mathbf{K}_0, \mathbf{c}_i; \text{stats}_i) + \epsilon$ {streaming computes $\mathbf{B}_i \mathbf{c}_i$}
11:    $\mathbf{Y}_i \leftarrow \texttt{FlashAttnApply}(\mathbf{Q}_i, \mathbf{K}_0, \mathbf{S}_i; \text{stats}_i)$ {streaming computes $\mathbf{B}_i \mathbf{S}_i$}
12:    $\mathbf{O}_i \leftarrow \mathbf{Y}_i \oslash \mathbf{r}_i[:, None]$ {$\widetilde{\mathbf{B}}_i \mathbf{S}_i$ without materializing $\widetilde{\mathbf{B}}_i$}
13: Reshape $\{\mathbf{O}_i\} \to \mathbf{O} \in \mathbb{R}^{L \times d}$ and **return**.

---

Therefore, $\mathbf{r}_i[s]$ is not an ad-hoc surrogate: it is the *explicit partition (row-sum) of the unnormalized video-group weights* induced by Echo, and $\widehat{\mathbf{W}}^{vv}$ is normalized by construction.

**Final joint output for video queries.**   Let $\mathbf{o}_{i,s}^{vid} \in \mathbb{R}^d$ denote the Echo output for this video query token (the within-video-group normalized output produced by *Echo-Row* or *Echo-Col*). Let $\log Z_{i,s}^{text}$ and $\mathbf{o}_{i,s}^{text}$ be computed exactly from text keys as in Eq. (36). We then compute the mixing coefficient using $\widehat{Z}_{i,s}^{vid} = \mathbf{r}_i[s]$, i.e., the video-group normalizer induced by Echo for the approximated video weights. This is not an unbiased estimator of the teacher's mixing coefficient $\lambda$; instead, it defines a normalized student attention whose mismatch to the full-attention teacher is corrected by distillation.

$$\lambda_{i,s} = \frac{\exp(\log Z_{i,s}^{text})}{\exp(\log Z_{i,s}^{text}) + \widehat{Z}_{i,s}^{vid}} = \sigma\Big(\log Z_{i,s}^{text} - \log \widehat{Z}_{i,s}^{vid}\Big), \tag{39}$$

$$\mathbf{o}_{i,s} = \lambda_{i,s}\, \mathbf{o}_{i,s}^{text} + (1 - \lambda_{i,s})\, \mathbf{o}_{i,s}^{vid}. \tag{40}$$

This preserves the joint-softmax competition between text and video groups exactly within our approximated attention, while keeping text interactions exact and leaving the EchoAttention operator unchanged.

**Exact text-query rows.**   For completeness, text queries (the top block-row in Eq. (32)) are computed exactly as standard attention over all keys:

$$\mathbf{W}^{text \to all} = \text{softmax}\Big(\mathbf{Q}^{text}\mathbf{K}^\top/\sqrt{d}\Big) \in \mathbb{R}^{L_{text} \times L}, \qquad \mathbf{O}^{text} = \mathbf{W}^{text \to all}\mathbf{V} \in \mathbb{R}^{L_{text} \times d}. \tag{41}$$

By construction, this yields $(\mathbf{W}^{tt}, \mathbf{W}^{tv})$ exactly.

**Complexity.**   Exact text-query attention (Eq. (41)) costs $O(L_{text} \cdot L \cdot d)$ and is typically small since $L_{text} \ll L$. For video queries, exact text-key computation costs $O(n\ell \cdot L_{text} \cdot d)$, while the video–video computation follows EchoAttention with dominant $O(n\ell^2 d)$ plus lower-order $O(n^2\ell d)$ terms. The mixing in Eqs. (39)–(40) adds only $O(n\ell)$ overhead. Thus, the core acceleration mechanism and complexity of EchoAttention remain unchanged under MM-DiT joint attention.

## B. Baseline Implementation Details

**VSA.** We implement Video Sparse Attention (VSA) (Zhang et al., 2025c) following its hierarchical two-stage design. Video tokens are partitioned into spatiotemporal cubes that map to hardware-friendly tiles; we use the standard tile size of 64 tokens (i.e., a $(4, 4, 4)$ cube) to align with the kernel layout. The *coarse stage* aggregates each cube into a single representation and

**Algorithm 4** EchoRowHeadBwd.

---

**Require:** $\nabla\mathbf{O} \in \mathbb{R}^{L\times d}$; $\mathbf{Q}, \mathbf{K}, \mathbf{V}$; intermediates $(\texttt{stats}, \mathbf{e}, \mathbf{r}, \mathbf{S})$; $\boldsymbol{\Theta}_q, \boldsymbol{\Theta}_k$.
**Ensure:** $\nabla\mathbf{Q}, \nabla\mathbf{K}, \nabla\mathbf{V}, \nabla\boldsymbol{\Theta}_q, \nabla\boldsymbol{\Theta}_k$.

1: Initialize all gradients to zero; reshape to frame-wise form $\{\mathbf{Q}_i, \mathbf{K}_i, \mathbf{V}_i\}$ and $\{\nabla\mathbf{O}_i\}$.
2: *Step 3 backprop (normalization and streaming apply):*
3: **for** $i = 0..n-1$ **do**
4:    $\nabla\mathbf{Y}_i \leftarrow \nabla\mathbf{O}_i \oslash \mathbf{r}_i[:, None]$
5:    $\nabla\mathbf{r}_i \leftarrow -\,\text{rowsum}(\nabla\mathbf{O}_i \odot \mathbf{Y}_i) \oslash \mathbf{r}_i$
6:    $(\nabla\mathbf{Q}_i^{(Y)}, \nabla\mathbf{K}_0^{(Y)}, \nabla\mathbf{S}_i) \leftarrow \texttt{FlashAttnBwd}(\mathbf{Q}_i, \mathbf{K}_0, \mathbf{S}_i, \nabla\mathbf{Y}_i; \texttt{stats}_i)$ {FlashAttention streaming}
7:    $(\nabla\mathbf{Q}_i^{(r)}, \nabla\mathbf{K}_0^{(r)}, \nabla\mathbf{c}_i) \leftarrow \texttt{FlashAttnBwd}(\mathbf{Q}_i, \mathbf{K}_0, \mathbf{c}_i, \nabla\mathbf{r}_i; \texttt{stats}_i)$ {FlashAttention streaming}
8:    $\nabla\mathbf{Q}_i \mathrel{+}= \nabla\mathbf{Q}_i^{(Y)} + \nabla\mathbf{Q}_i^{(r)}$
9:    $\nabla\mathbf{K}_0 \mathrel{+}= \nabla\mathbf{K}_0^{(Y)} + \nabla\mathbf{K}_0^{(r)}$
10: **end for**
11: *Step 2 backprop (diagonal mixing and projections):*
12:    From $\mathbf{S}_i[s] = \sum_j \mathbf{e}_{i,j}[s]\mathbf{V}_j[s]$:
13:    $\nabla\mathbf{V}_j[s] \mathrel{+}= \sum_i \mathbf{e}_{i,j}[s] \cdot \nabla\mathbf{S}_i[s];\quad \nabla\mathbf{e}_{i,j}[s] \mathrel{+}= \langle\nabla\mathbf{S}_i[s], \mathbf{V}_j[s]\rangle + \nabla\mathbf{c}_i[s]$
14:    $\nabla\boldsymbol{\alpha}_{i,j} \leftarrow \nabla\mathbf{e}_{i,j} \odot \mathbf{e}_{i,j};\quad \nabla\mathbf{Z}_j \mathrel{+}= \sum_i \nabla\boldsymbol{\alpha}_{i,j}[:, None]\mathbf{u}_i[None, :]/\sqrt{d_{proj}};\quad \nabla\mathbf{u}_i \mathrel{+}= \sum_j \mathbf{Z}_j^\top \nabla\boldsymbol{\alpha}_{i,j}/\sqrt{d_{proj}}$
15:    $\nabla\boldsymbol{\Theta}_k \mathrel{+}= \sum_j (\nabla\mathbf{Z}_j)^\top \mathbf{K}_j;\quad \nabla\mathbf{K}_j \mathrel{+}= \nabla\mathbf{Z}_j\boldsymbol{\Theta}_k$
16:    $\nabla\bar{\mathbf{q}}_i \mathrel{+}= \boldsymbol{\Theta}_q^\top \nabla\mathbf{u}_i;\quad \nabla\boldsymbol{\Theta}_q \mathrel{+}= \sum_i \nabla\mathbf{u}_i\,\bar{\mathbf{q}}_i^\top;\quad \nabla\mathbf{Q}_i[s] \mathrel{+}= \nabla\bar{\mathbf{q}}_i/\ell$
17: *Step 1 backprop:* included in the two $\texttt{FlashAttnBwd}$ calls above (streaming softmax backward using $\texttt{stats}_i$).
18: Reshape back and **return**.

---

*Table 7.* VSA budget settings. $m = L/64$ is the number of KV tiles. "average top-$k$" reports the average retained token-pair ratio implied by the chosen $K$.

|  | Wan2.1-1.3B | CogVideoX1.5-5B |
|---|---|---|
| Tile size | 64 | 64 |
| $K$ (top tiles per query tile) | 52 | 106 |
| Avg. top-$k$ | 0.102 | 0.151 |

performs dense cube-to-cube attention to identify salient regions. The *fine stage* then computes token-level attention only within the top-$K$ selected key cubes for each query cube using a block-sparse kernel. The final attention output is a gated mixture of coarse and fine outputs, as in (Zhang et al., 2025c). We set $K$ (and the corresponding token-pair retention ratio top-$k$) as summarized in Table 7. All other hyperparameters and execution details follow the default settings in the original method. We also train it for 3000 steps with a batch size of 64.

**SLA.** We implement SLA (Zhang et al., 2025a) using its default configuration and kernel settings. We set the sparsity knob to top-$k = 0.1$ for Wan2.1-1.3B and top-$k = 0.15$ for CogVideoX1.5-5B. All other hyperparameters and execution details in SLA follow the defaults in the original method. We also train it for 3000 steps with a batch size of 64.

**SpargeAttn (trainable variant).** SpargeAttn (Zhang et al., 2025b) is originally a training-free method that predicts a block-sparse mask online using block-mean query/key statistics and a cumulative-probability selector (TopCdf). Its key control variable is a threshold $\tau$ (equivalently, a *recall* target), which determines the speed–quality trade-off (and yields an implicit average top-$k$). The method uses an auxiliary hyperparameter $L1$ to grid-search $\tau$ for each head.

For a fair post-training comparison under our distillation setting, we keep the default SpargeAttn pipeline and tune only $L1$, which indirectly determines the selected $\tau$ values and thus the resulting sparsity. This yields two variants: *SpargeAttn-Dense* and *SpargeAttn-Sparse*. Their $L1$ settings and the resulting averaged statistics (mean $\tau$ (recall) and average top-$k$) are reported in Table 8. After fixing $\tau$ by the above procedure, we distill a student that follows the SpargeAttn attention computation[3] from a full-attention teacher using the same objective in Eq. (20). All other training sets follow as in §4.3 and Appendix C.

---

[3]We disable attention quantization in SpargeAttn to ensure a fair comparison.

---

**Algorithm 5** `EchoColHeadFwd`.

---

**Require:** $\mathbf{Q}, \mathbf{K}, \mathbf{V} \in \mathbb{R}^{L \times d}$; $\ell$ $(n = L/\ell)$; $\boldsymbol{\Theta}_q, \boldsymbol{\Theta}_k$.
**Ensure:** $\mathbf{O} \in \mathbb{R}^{L \times d}$ and intermediates (`stats`, $\mathbf{T}, \mathbf{e}, \mathbf{r}$).
1: Reshape $\mathbf{Q}, \mathbf{K}, \mathbf{V} \rightarrow \{\mathbf{Q}_i, \mathbf{K}_j, \mathbf{V}_j\}_{i,j=0}^{n-1}$ with $\mathbf{Q}_i, \mathbf{K}_j, \mathbf{V}_j \in \mathbb{R}^{\ell \times d}$.
2: *Step 1 (column prototypes, FlashAttention streaming):*
3: **for** $j = 0..n-1$ **do**
4:     $\text{stats}_j \leftarrow \texttt{FlashAttnStats}(\mathbf{Q}_0, \mathbf{K}_j)$ {streaming; no materialization of $\mathbf{B}_j$}
5:     $\mathbf{T}_j \leftarrow \texttt{FlashAttnApply}(\mathbf{Q}_0, \mathbf{K}_j, \mathbf{V}_j; \text{stats}_j)$ {streaming computes $\mathbf{B}_j \mathbf{V}_j$}
6: **end for**
7: *Step 2:* Query-aligned diagonal weights and normalization:
8:     $\mathbf{Z}_i \leftarrow \mathbf{Q}_i \boldsymbol{\Theta}_q \in \mathbb{R}^{\ell \times d_{proj}}$;    $\bar{\mathbf{k}}_j \leftarrow \text{mean}_s(\mathbf{K}_j) \in \mathbb{R}^d$;    $\mathbf{u}_j \leftarrow \boldsymbol{\Theta}_k \bar{\mathbf{k}}_j \in \mathbb{R}^{d_{proj}}$
9:     $\mathbf{e}_{i,j} \leftarrow \exp\!\big(\mathbf{Z}_i \mathbf{u}_j / \sqrt{d_{proj}}\big) \in \mathbb{R}_+^{\ell}$;    $\mathbf{r}_i \leftarrow \sum_j \mathbf{e}_{i,j} \in \mathbb{R}_+^{\ell}$
10: *Step 3:* Left-diagonal mixing:
11:     $\mathbf{O}_i[s] \leftarrow \sum_j \frac{\mathbf{e}_{i,j}[s]}{\mathbf{r}_i[s] + \epsilon} \cdot \mathbf{T}_j[s] \in \mathbb{R}^d$.
12: Reshape $\{\mathbf{O}_i\} \rightarrow \mathbf{O} \in \mathbb{R}^{L \times d}$ and **return**.

---

*Table 8.* SpargeAttn settings and resulting averaged sparsity statistics under our protocol. "Average top-$k$" denotes the average retained token-pair ratio induced by the learned/selected masks.

| Model | Variant | $L1$ | Mean $\tau$ (recall) | Avg. top-$k$ |
|---|---|---|---|---|
| Wan2.1-1.3B | SpargeAttn-Dense | 0.06 | 0.732 | 0.201 |
| | SpargeAttn-Sparse | 0.09 | 0.596 | 0.092 |
| CogVideoX1.5-5B | SpargeAttn-Dense | 0.08 | 0.792 | 0.285 |
| | SpargeAttn-Sparse | 0.15 | 0.639 | 0.142 |

**Sparse-only.** This ablation removes the router and the Echo branch entirely and applies our *Sparse OP* to *all* heads at every timestep and layer (with the same default top-$k$ used in the main experiments for each backbone). Since no routing is involved, we distill this model under deterministic (hard) operator choices for 3000 steps with a batch size of 64 using the same distillation loss, LoRA setup and other training setups as in §4.3 and Appendix C.

**Echo-only.** This ablation removes the router and the Sparse branch and applies the *Echo* operator to *all* heads. We still train $g_{\text{row}}$ to select between *Echo-Row* and *Echo-Col*. We run the full three-stage schedule and distill using the same objective, LoRA setup and other training setups as in §4.3 and Appendix C.

**Echo-full.** This baseline isolates the effect of routing from the *Echo* operator itself. We first take the hard routing table produced by a fully trained EchoAttention model from our main experiments and keep this table fixed. We then replace every head routed to Echo with *full attention*, while keeping heads routed to Sparse unchanged. Under this fixed routing, we distill for 3000 steps with a batch size of 64 using the same distillation loss, LoRA setup and other training setups as in §4.3 and Appendix C.

**Manual-routing.** This baseline replaces learned routing with an offline handcrafted routing table. Following the spirit of Observation 3, we estimate, for each (timestep, layer, head) triple, which operator best approximates the teacher by comparing approximation errors on a small calibration set. Concretely, we sample 64 training examples, run the teacher attention to obtain reference outputs, and evaluate the three candidate operators (Sparse with the default top-$k$, *Echo-Row*, and *Echo-Col*) on the same inputs. We select the operator that minimizes the mean approximation error (averaged over the calibration set) and record the resulting deterministic lookup table. We then distill the student for 3000 steps with a batch size of 64 under this fixed hard routing table (no gate learning), using the same loss, LoRA setup and other training setups as in §4.3 and Appendix C.

**Top-1 Similarity prototype.** This ablation replaces the fixed 0-index prototype in *Echo-Row/Echo-Col* with an adaptive choice based on a frame-level similarity score computed from low-dimensional frame summaries. We define

$$\bar{\mathbf{q}}_i = \tfrac{1}{\ell} \sum_{s=0}^{\ell-1} \mathbf{Q}_i[s] \in \mathbb{R}^d, \qquad \bar{\mathbf{k}}_j = \tfrac{1}{\ell} \sum_{s=0}^{\ell-1} \mathbf{K}_j[s] \in \mathbb{R}^d,$$

---

**Algorithm 6** `EchoColHeadBwd`.

---

**Require:** $\nabla \mathbf{O} \in \mathbb{R}^{L \times d}$; $\mathbf{Q}, \mathbf{K}, \mathbf{V}$; intermediates $(\texttt{stats}, \mathbf{T}, \mathbf{e}, \mathbf{r})$; $\boldsymbol{\Theta}_q, \boldsymbol{\Theta}_k$.
**Ensure:** $\nabla \mathbf{Q}, \nabla \mathbf{K}, \nabla \mathbf{V}, \nabla \boldsymbol{\Theta}_q, \nabla \boldsymbol{\Theta}_k$.

1: Initialize all gradients to zero; reshape to frame-wise form.
2: Define normalized weights $\mathbf{w}_{i,j} \triangleq \mathbf{e}_{i,j} \oslash (\mathbf{r}_i + \epsilon)$ (elementwise).
3: *Step 3 backprop (mixing):*
4:    $\nabla \mathbf{T}_j[s] \mathrel{+}= \sum_i \mathbf{w}_{i,j}[s] \cdot \nabla \mathbf{O}_i[s]$.
5:    $\nabla \mathbf{w}_{i,j}[s] \mathrel{+}= \langle \nabla \mathbf{O}_i[s], \mathbf{T}_j[s] \rangle$.
6: *Step 2 backprop (normalization and exp):*
7:    Using $\mathbf{w}_{i,j} = \mathbf{e}_{i,j}/(\mathbf{r}_i + \epsilon)$ and $\mathbf{r}_i = \sum_j \mathbf{e}_{i,j}$, accumulate $\nabla \mathbf{e}_{i,j}$ from $\nabla \mathbf{w}_{i,j}$ (standard softmax-like normalization over $j$ for each $i, s$).
8:    $\nabla \boldsymbol{\alpha}_{i,j} \leftarrow \nabla \mathbf{e}_{i,j} \odot \mathbf{e}_{i,j} \; \{ \mathbf{e}_{i,j} = \exp(\boldsymbol{\alpha}_{i,j}) \}$
9:    Let $\boldsymbol{\alpha}_{i,j} = \mathbf{Z}_i \mathbf{u}_j / \sqrt{d_{proj}}$. Then
10:    $\nabla \mathbf{Z}_i \mathrel{+}= \sum_j \nabla \boldsymbol{\alpha}_{i,j}[:, None] \mathbf{u}_j[None, :]/\sqrt{d_{proj}}, \quad \nabla \mathbf{u}_j \mathrel{+}= \sum_i \mathbf{Z}_i^\top \nabla \boldsymbol{\alpha}_{i,j}/\sqrt{d_{proj}}$.
11:    $\nabla \boldsymbol{\Theta}_q \mathrel{+}= \sum_i (\nabla \mathbf{Z}_i)^\top \mathbf{Q}_i; \quad \nabla \mathbf{Q}_i \mathrel{+}= \nabla \mathbf{Z}_i \boldsymbol{\Theta}_q$.
12:    $\nabla \bar{\mathbf{k}}_j \mathrel{+}= \boldsymbol{\Theta}_k^\top \nabla \mathbf{u}_j; \quad \nabla \boldsymbol{\Theta}_k \mathrel{+}= \sum_j \nabla \mathbf{u}_j \bar{\mathbf{k}}_j^\top$.
13:    Distribute mean: $\nabla \mathbf{K}_j[s] \mathrel{+}= \nabla \bar{\mathbf{k}}_j/\ell$ for all $s$.
14: *Step 1 backprop (prototype apply, FlashAttention streaming):*
15: **for** $j = 0..n-1$ **do**
16:    $(\nabla \mathbf{Q}_0^{(j)}, \nabla \mathbf{K}_j^{(T)}, \nabla \mathbf{V}_j) \leftarrow \texttt{FlashAttnBwd}(\mathbf{Q}_0, \mathbf{K}_j, \mathbf{V}_j, \nabla \mathbf{T}_j; \texttt{stats}_j) \; \{\text{streaming softmax backward}\}$
17:    $\nabla \mathbf{Q}_0 \mathrel{+}= \nabla \mathbf{Q}_0^{(j)}; \quad \nabla \mathbf{K}_j \mathrel{+}= \nabla \mathbf{K}_j^{(T)}$
18: **end for**
19: Reshape back and **return**.

---

$$\mathbf{u}_i = \boldsymbol{\Theta}_q \bar{\mathbf{q}}_i \in \mathbb{R}^{d_{\mathrm{proj}}}, \qquad \mathbf{v}_j = \boldsymbol{\Theta}_k \bar{\mathbf{k}}_j \in \mathbb{R}^{d_{\mathrm{proj}}}.$$

For *Echo-Row*, for each query frame $i$ we select

$$j^*(i) = \arg \max_{j \in \{0, \dots, n-1\}} \langle \mathbf{u}_i, \mathbf{v}_j \rangle, \qquad \mathbf{B}_i = \mathrm{softmax}\Big(\mathbf{Q}_i \mathbf{K}_{j^*(i)}^\top / \sqrt{d}\Big) \in \mathbb{R}^{\ell \times \ell},$$

and use $\mathbf{B}_i$ as the row prototype in Step 1 (replacing Eq. (12)). For *Echo-Col*, for each key frame $j$ we select

$$i^*(j) = \arg \max_{i \in \{0, \dots, n-1\}} \langle \mathbf{u}_i, \mathbf{v}_j \rangle, \qquad \mathbf{B}_j = \mathrm{softmax}\Big(\mathbf{Q}_{i^*(j)} \mathbf{K}_j^\top / \sqrt{d}\Big) \in \mathbb{R}^{\ell \times \ell},$$

and use $\mathbf{B}_j$ as the column prototype in Step 1 (replacing Eq. (23)). All other components (routing, calibration in Eq. (14)/Eq. (25), and distillation setup) are kept identical to the main method as in §4.3 and Appendix C.

## C. Other Distillation Details

We distill an EchoAttention student from the original full-attention teacher by matching the teacher's noise prediction (Eq. (20)). The teacher is kept frozen throughout training. The student is initialized from the teacher weights, where we replace each full-attention module with our routed EchoAttention module (*Sparse* / *Echo-Row* / *Echo-Col*) and keep all other components identical.

**Data Processing.** Our dataset is collected from public sources and curated through manual and automated cleaning. It covers a wide range of video resolutions and durations. During training, we resize each training video to the target resolution and duration used by the corresponding model.

**EchoAttention Parameters and initialization.** For the new parameters introduced by EchoAttention, we train (1) the routing parameters that produce $g_{\mathrm{sim}}(t, h)$ and $g_{\mathrm{row}}(t, h)$ at each layer, which are initialized to zero; and (2) the low-dimensional projection modules $\boldsymbol{\Theta}_q$ and $\boldsymbol{\Theta}_k$ used by *Echo OP*, which are initialized with Xavier uniform initialization.

**LoRA setup.** By default, we apply LoRA to all linear layers in the student model's attention and FFN blocks. We also train the head layers as well as the `text_embedding` and `time_embedding` layers. Unless otherwise specified, we use LoRA rank $r = 128$ and set `lora_alpha=128`.

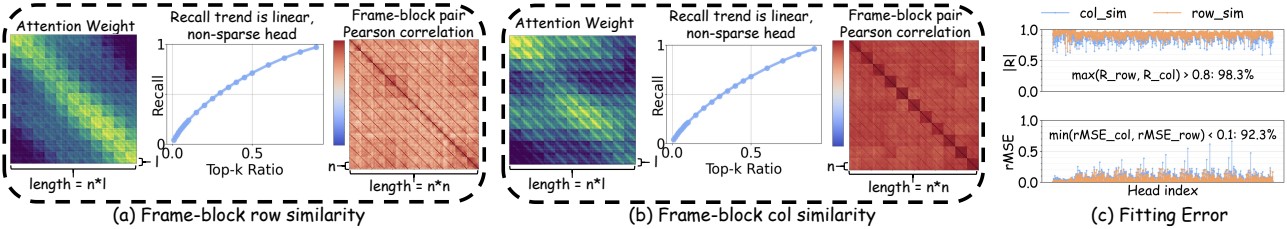

Figure 13. *Frame-block similarity* visualization on CogVideoX1.5-5B.

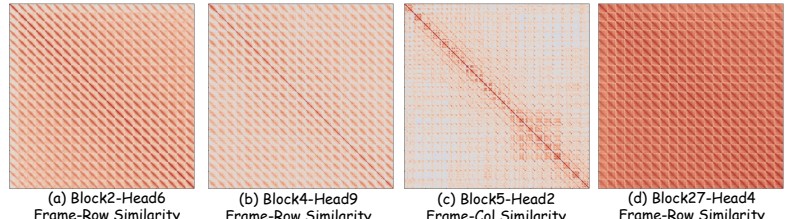

Figure 14. *Frame-block similarity* Pearson correlation coefficient examples on Wan2.1-1.3B.

Table 9. Scalability on *Wan2.1-1.3B* with 10-second 480p generation.

| Method | VBench↑ | PSNR↑ | SSIM↑ | LPIPS↓ | Sparsity | Latency↓ | Speedup↑ |
|---|---|---|---|---|---|---|---|
| Full Attention | 80.2 | - | - | - | - | 514 | 1.00× |
| SLA | 79.5 | 18.32 | 0.6910 | 0.2878 | 90% | 213 | 2.41× |
| **Ours (EchoAttention)** | **80.1** | **19.95** | **0.7424** | **0.2328** | **95%** | **180** | **2.86×** |

Table 10. Compatibility with AR+DiT backbones. Self-Forcing is evaluated on 5-second 480p generation and LongLive on 60-second 480p generation.

| Model | Method | VBench↑ | PSNR↑ | SSIM↑ | LPIPS↓ | Sparsity | Latency↓ | Speedup↑ |
|---|---|---|---|---|---|---|---|---|
| | Full Attention | 83.9 | - | - | - | - | 12.5 | 1.00× |
| Self-Forcing | SLA | 83.3 | 18.62 | 0.6951 | 0.2929 | 80% | 10.6 | 1.18× |
| | **Ours (EchoAttention)** | **84.0** | **20.10** | **0.7324** | **0.2306** | **85%** | **9.8** | **1.28×** |
| | Full Attention | 83.3 | - | - | - | - | 91.0 | 1.00× |
| LongLive | SLA | 82.6 | 17.86 | 0.6693 | 0.3194 | 80% | 66.8 | 1.36× |
| | **Ours (EchoAttention)** | **83.2** | **18.76** | **0.6885** | **0.2917** | **85%** | **61.2** | **1.49×** |

**Optimization and numerics.** We use AdamW with learning rate $1 \times 10^{-4}$, $(\beta_1, \beta_2) = (0.9, 0.999)$, and weight decay 0.01. For the routing gates $g_{\text{sim}}(t, h)$ and $g_{\text{row}}(t, h)$, we use a larger learning rate of $1 \times 10^{-1}$ since these parameters converge more slowly. Training is performed with mixed precision (bf16) and global-norm gradient clipping at 1000.0.

# D. Scalability and Compatibility

To evaluate scalability, we further test *Wan2.1-1.3B* on longer video generation. Specifically, we generate 10-second 480p videos and report the speed–quality metrics in Table 9. Overall, EchoAttention maintains a better speed–quality trade-off than the strongest baseline SLA at longer generation length.

We also apply EchoAttention to AR+DiT backbones, including Self-Forcing (Huang et al., 2026) and LongLive (Yang et al., 2025a), as shown in Table 10. These models use chunked autoregressive generation with step-wise distillation and KV caching, which reduces the fraction of total latency spent in attention. Therefore, the absolute speedup from attention-level acceleration is smaller than on standard DiTs, but EchoAttention still improves the speed-quality trade-off over SLA.

We further note that recent accelerated video generation backbones based on step distillation or few-step inference, such as FastHunyuan (FastVideo, 2025) and FastWan (FastDM, 2025), reduce the number of denoising steps, which is orthogonal to EchoAttention's attention-level acceleration within each step.

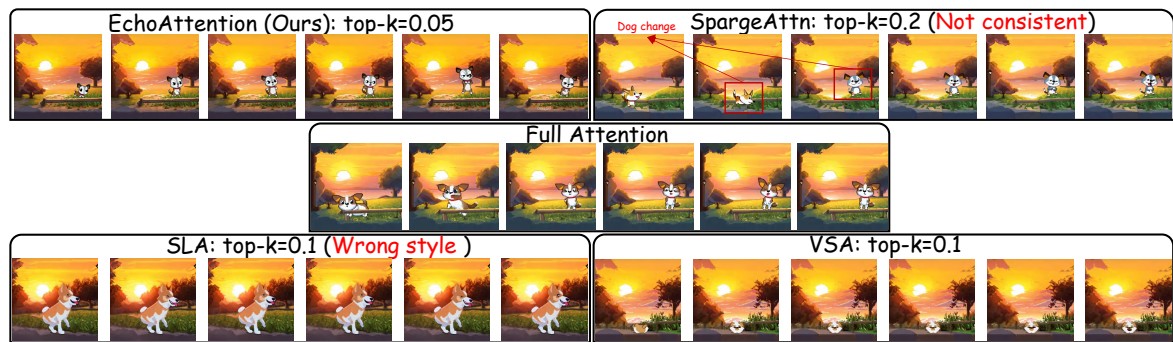

*Figure 15. Frame-block similarity* Pearson correlation coefficient examples on CogVideoX1.5-5B.

## E. More Observational Proofs

In the main text, we primarily use Wan2.1-1.3B to illustrate our observations for clarity. Here, we provide additional evidence on the MM-DiT backbones, CogVideoX1.5-5B.

For CogVideoX1.5-5B, which uses joint attention over concatenated text and video tokens, we apply the analysis to the video–video sub-block $\mathbf{W}^{vv}$ (Eq. (33)) and keep text-related interactions unchanged, consistent with Appendix A.3. Figure 13 shows that the video block $\mathbf{W}^{vv}$ also presents clear frame-block similarity patterns along block-rows/columns, and the same diagonal calibration provides an effective low-cost approximation on non-sparse heads.

We also provide more *frame-block similarity* Pearson correlation coefficient examples in Figures 14 and 15.

## F. More Visual Examples and Limitations

We provide additional qualitative comparisons to complement the visual examples in §5.4. Figures 16–19 present additional results under the same evaluation protocol, showing that EchoAttention better preserves fine-grained motion coherence and appearance consistency under aggressive acceleration.

To better understand the boundary of our method, we also include a failure case in Figure 20. In this example, both our method and the original full attention model exhibit a hallucination, producing an implausible hybrid where the cat's head is fused onto a bird's body. The fact that this artifact appears in both outputs may suggest that distillation-based attention approximation can inherit undesirable behaviors already present in the full-attention teacher. However, such failures are rare in our experiments. In the vast majority of cases, our method yields higher visual quality than the SOTA baselines. We leave a systematic study of these failure modes, the precise conditions under which they arise, and potential mitigation strategies to future work.

*Figure 16.* `A cute happy Corgi playing in park, sunset, pixel art.`

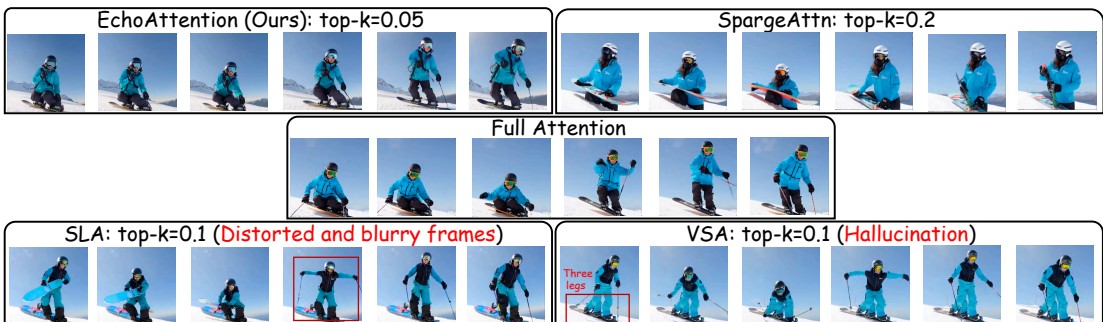

Figure 17. *Skis on the bottom of a snowboard, front view.*

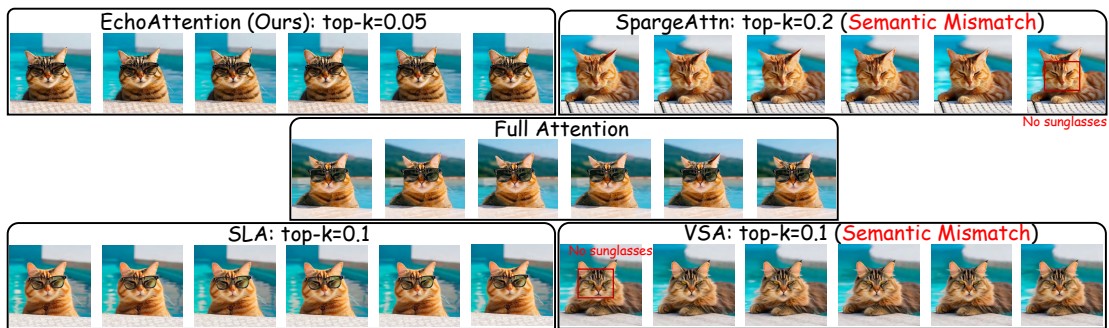

Figure 18. *A cat wearing sunglasses at a pool.*

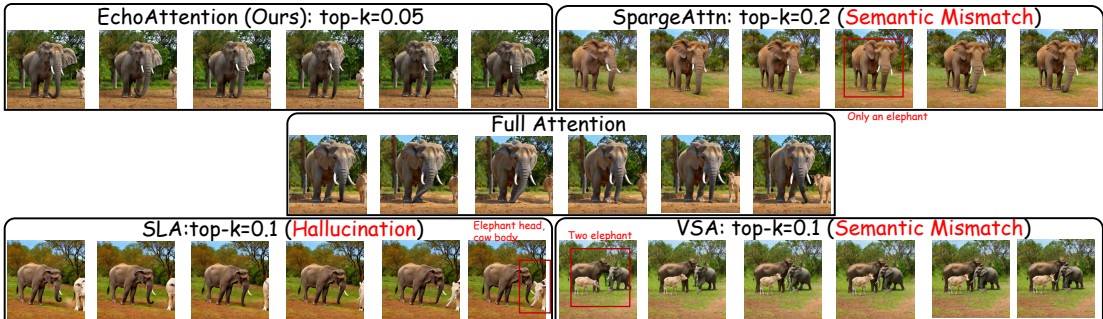

Figure 19. *A cow and an elephant.*

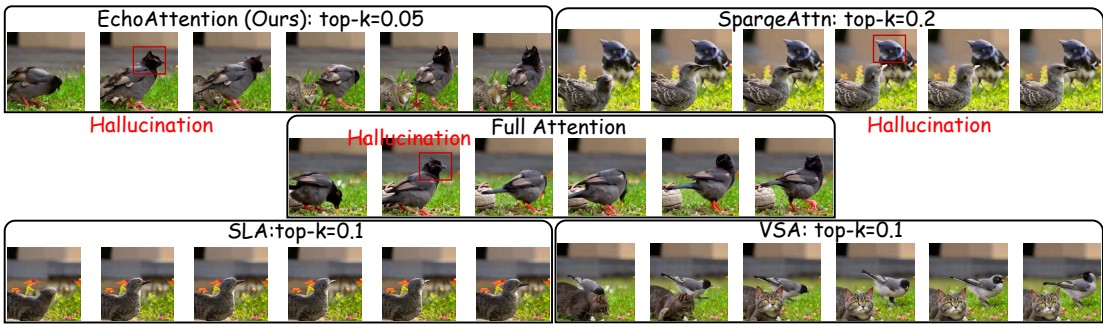

Figure 20. *A bird and a cat.*

