# OpenReview forum: "EchoAttention: Exploiting Token-Pair Redundancy and Frame-Block Similarity for Efficient Video Generation"
_ICML.cc/2026/Conference — ICML 2026 regular_

### Official Review · Reviewer_frpJ · 2026-02-18

**Soundness:** 3
**Presentation:** 2
**Significance:** 2
**Originality:** 2
**Overall Recommendation:** 4
**Confidence:** 4

**Summary:**

The authors introduce EchoAttention, a mechanism designed to optimize attention heads that remain computationally expensive under sparse attention frameworks within long-video Diffusion Transformers (DiTs). The paper proposes a three-stage distillation strategy that facilitates routing between EchoAttention and sparse attention modules. Empirical results demonstrate a speedup of up to 2.42x in latency while maintaining generation quality.

**Compliance With Llm Reviewing Policy:**

Affirmed.

**Final Justification:**

After reading the other authors review and rebuttal comments, I maintain my opinion as in my acknowledgement.

**Key Questions For Authors:**

Please refer to the weakness section.

1. Why is a two-level hierarchical gate ($g_{sim}$ and $g_{row}$) preferred over a simpler one-level, three-way gate?
2. Table 5 evaluates different schedules, but what is the intuition behind the specific choices of $soft=0.05$ and $st=0.8$? A fine-grained search showing the trend of these boundaries would provide better insight for future distillation-based routing.


 I would like to raise my recommendations if the authors can address the concerns.

**Limitations:**

Limitations and "bad cases" (such as the semantic/orientation shifting in the supplementary material `an airplane soaring through a clear blue sky` video) are not discussed in the main manuscript. Figure 20 acknowledges a hallucination, but more rigorous analysis of these more limitations is needed.

**Strengths And Weaknesses:**

# Strengths

1. The paper provides a compelling characterization of head-level heterogeneity, identifying that non-sparse heads, which typically govern the speed-quality floor, can be approximated through frame-block prototypes.
2. The Echo operator (Echo-Row and Echo-Col) is well-motivated, specifically reducing cost for high-impact, non-sparse heads.
3. The manuscript is logically structured, and the transition from motivating observations to the architectural solution is smooth.
4. The paper’s logic is smooth and clear, the proposed method is well presentated.
5. The authors include detailed ablations on prototype selection and operator suitability, which clarify the contributions of the individual components.

# Weakness

1. Table 4 shows that the learned routing policy (`Ours`, 83.2 VBench) provides only marginal improvements over a handcrafted `manual-rt` (82.9 VBench). Given the high cost of a three-stage distillation process, the necessity of such a complex training-based routing policy is not fully justified.
2. Current SOTA for video generation often relies on step-distillation to reduce sampling steps (e.g., 2-6 steps) [1-4]. The paper lacks analysis on how operation-level optimization like EchoAttention interacts with these sampling-level accelerations, raising concerns about its practical significance in modern inference pipelines.
3. Although the title emphasizes "Long Video Generation," most of the main manuscript focuses on 5-second sequences. In the current landscape of video DiTs, 5 seconds is considered standard duration rather than "long". The scalability test to 10 seconds is only mentioned in one section. Maybe renaming makes the title and content more aligned.

# Minor weakness

1. The differences between EchoAttention and baselines like SLA, VSA, or SpargeAttn are not sufficiently highlighted in the Related Work section. More detail here would better emphasize the novelty of targeting frame-block similarity over simple linear compensation.
2. Several equations are nearly identical (e.g., Eq. 3 and Eq. 6 for sparse attention; Eq. 11 and Eq. 18 for Echo-Row). Consolidating these would improve readability. And the distinction between training-only components and inference-only kernels in Section 4.1 is occasionally blurred.
3. While Appendix provides pseudocode, the absence of ready-to-run source code or pre-trained weights limits the verification of the 2.42x speedup claims on diverse hardware.

# Reference
1. Yin, Tianwei, et al. "One-step diffusion with distribution matching distillation. 2024 IEEE." CVF Conference on Computer Vision and Pattern Recognition (CVPR). 2023.
2. Wang, Fu-Yun, et al. "Phased consistency models." Advances in neural information processing systems 37 (2024): 83951-84009.
3. https://huggingface.co/FastVideo/FastHunyuan
4. https://huggingface.co/FastDM/Wan2.2-T2V-A14B-Merge-Lightning-V1.0-Diffusers

---

> ### Author Rebuttal · Authors · 2026-03-30
>
> We sincerely thank the reviewer for the constructive feedback. We address each concern below.
>
> **frpJ-Q1: Learned vs. manual routing.** We appreciate the question.
>
> First, regarding the significance of 0.3 VBench improvement: on the VBench leaderboard, Open-Sora 2.0 vs. Sora differ by only 0.06, and dedicated enhancement methods like Enhance-A-Video achieve only +0.07 to +0.12. Our 0.3 gain from an *efficiency* technique is therefore non-trivial.
>
> Second, practically, the router is just two linear embeddings (~2% of block compute), and becomes an O(1) lookup at inference. It is co-trained with Echo OP and LoRA parameters, adding virtually no training time. In contrast, manual routing requires hours of offline profiling on carefully selected data. After that, separate training for Echo OP and LoRA is still needed, making the pipeline non-e2e and arguably more complex. We will add a discussion on it in the revised paper.
>
> **frpJ-Q2: Step-distillation compatibility.** Thank you for the question. We verified frame-block similarity on step-distilled models, including FastHunyuan, Wan2.2-T2V-A14B-Merge-Lightning-V1.0, and FastWan2.1-T2V-1.3B (https://anonymous.4open.science/r/ICML26_367_rebuttal_figures-A789 Fig1–3), and all exhibit clear frame-block redundancy. Due to rebuttal time constraints, we only trained on a relatively small 3-step model FastWan2.1-T2V-1.3B (https://github.com/hao-ai-lab/FastVideo):
>
> |Method|VBench|PSNR|SSIM|LPIPS|Sparsity|Lat.|Spd.|
> |---|--:|--:|--:|--:|--:|--:|--:|
> |Full Attn|81.9|-|-|-|-|12.3|1.00|
> |SLA|81.2|16.86|0.6705|0.2954|90%|6.95|1.77|
> |**Ours**|**82.1**|**18.21**|**0.7211**|**0.2571**|**95%**|**6.41**|**1.91**|
>
> This confirms that operator-level and sampling-level accelerations are complementary. We will cite the provided references and include these results.
>
> **frpJ-Q3: Title "Long Video".** We appreciate the concern. We agree that 5s is standard rather than "long". Our method is effective for both standard and long video generation (see 60s LongLive in **fKQd-Q2**), and we will revise the title to remove "Long" to reflect its applicability.
>
> **frpJ-Q4: Related work.** Thank you for the suggestion. We note in the Introduction (the Limitation part) the key differences, including redundancy type and handling of non-sparse heads. Regarding the concern about SLA’s simple linear compensation, we clarify that SLA is built around sparse heads, and for non-sparse heads where information loss is too severe, linear compensation is insufficient. Our Echo OP instead exploits structural frame-block similarity, preserving the attention structure rather than patching a sparse approximation. We will reorganize Section 2 to highlight these points more clearly.
>
> **frpJ-Q5 & Q9: Equations & limitations.** We will consolidate Eq.3/6 and Eq.11/18 and add clear section markers. We will also add a limitations paragraph covering semantic drift (Figure 20) and a post-training-only scope.
>
> **frpJ-Q6: Source code & hardware.** We will release all code upon acceptance. To address concerns about hardware generality, we tested on NVIDIA H20 and L20 GPUs:
>
> |GPU|Method|Wan Lat. (s)|Wan Spd.|Cog Lat. (s)|Cog Spd.|
> |---|---|---:|---:|---:|---:|
> |H20|Full Attn|246|1.00|1182|1.00|
> ||SLA|132|1.86|560|2.11|
> ||**Ours**|**120**|**2.05**|**522**|**2.26**|
> |L20|Full Attn|366|1.00|1523|1.00|
> ||SLA|258|1.42|842|1.81|
> ||**Ours**|**238**|**1.54**|**800**|**1.90**|
>
> While absolute speedup ratios vary across hardware, EchoAttn consistently achieves the best speed–quality Pareto frontier, showing the efficiency advantage generalizes across GPUs.
>
> **frpJ-Q7: Two or one level gate.** We appreciate the question. On one hand, the routing decision is naturally hierarchical: the first level chooses between fundamentally different OP types (Sparse vs. Echo), while the second selects between variants within the Echo family (Row vs. Col). On the other hand, binary decisions via the straight-through estimator are more stable than three-way discrete routing (arXiv:1308.3432). Empirically (one-level 3-way = 3-class Gumbel-Softmax with straight-through):
>
> |Method|VBench|PSNR|SSIM|LPIPS|
> |---|--:|--:|--:|--:|
> |One-level 3-way|82.6|18.04|0.6896|0.3091|
> |**Ours (2-level)**|**83.2**|**19.38**|**0.7121**|**0.2661**|
>
> The two-level design outperforms by +0.6 VBench and +1.34 PSNR.
>
> **frpJ-Q8: Schedule boundaries.** Beyond Table 5 and 6, we conducted a fine-grained sweep (see **mPrC-Q2**) to validate robustness. The sweep shows insensitivity across non-extreme settings.
>
> - For soft=0.05: a short soft phase suffices to warm up routing and prevent premature collapse, while larger values (0.1, 0.2) yield comparable results but excessively large values (0.5) delay discrete routing.
> - For st=0.8: 20% hard stage suffices to finalize discrete decisions and eliminate train-inference mismatch, while st=0.95 compresses hard stage and causes quality drop (VBench 82.8, PSNR 18.67).
>
> Overall, (0.05, 0.8) is a reasonable choice within the insensitive range.

---

> > ### Author Rebuttal · Reviewer_frpJ · 2026-04-01
> >
> > I thank the authors for their comprehensive rebuttal. The newly provided ablation studies regarding the hyperparameter choices and the two-level gate design are insightful and effectively address my primary concerns. Furthermore, the demonstrated compatibility with few-step video models provides strong evidence for the generalizability of EchoAttention.
> >
> > I encourage the authors to incorporate these new results and address the remaining suggestions from other reviewers in the final version. I will raise my recommendation.

---

> > > ### Author Response · Authors · 2026-04-03
> > >
> > > We sincerely thank the reviewer for the constructive and insightful feedback throughout this process, and for raising the recommendation.
> > >
> > > We are glad that the new ablation studies and compatibility experiments have addressed the reviewer's concerns. We will incorporate all new results and address the remaining suggestions in the final version.

---

### Official Review · Reviewer_mPrC · 2026-03-09

**Soundness:** 3
**Presentation:** 3
**Significance:** 4
**Originality:** 3
**Overall Recommendation:** 4
**Confidence:** 2

**Summary:**

This paper proposes a structured attention acceleration framework for long video generation, called EchoAttention. The authors observe that, besides token-pair redundancy, video DiTs also exhibit strong frame-block similarity across attention weights. Experiments on Wan2.1-1.3B and CogVideoX1.5-5B show that the method achieves better speed-quality trade-offs than existing sparse attention methods.

**Compliance With Llm Reviewing Policy:**

Affirmed.

**Final Justification:**

The authors solved all my concerns in the rebuttal, so I maintain a positive score.

**Key Questions For Authors:**

1. How well would EchoAttention generalize to other video diffusion models or to different architectures such as autoregressive video generators?

2. Please discuss the computational overhead or sensitivity in deployment. While the paper reports latency improvements, could the authors provide the computational overhead introduced by the routing mechanism, and how much do they contirbute to the overall lantency? And how sensitive is the learned routing policy to its hyperparameters?

3. How is the temporal consistency of the generated videos?

4. Why can a 5-second video be classified as the long video generation task? Could the authors provide more extensive evaluations on longer video durations?

**Limitations:**

Yes

**Strengths And Weaknesses:**

Strengths:

1. The observation in this paper is insightful and motivates the design of the proposed framework.

2. The proposed dual-operator architecture with learned routing is well-motivated and supported by empirical studies and theory.

3. Experiments on two video diffusion models over seven metrics are comprehensive, and demonstrate the consistent improvements regarding both quality and speed.

Weaknesses:

1. The empirical evaluation is limited to two video diffusion models, and it is unclear how well the method generalizes to other architectures.

2. The paper reports end-to-end latency and speedup, which shows improvement compared with baselines. However, the routing mechanism adds architectural complexity and additional hyperparameters, yet the paper does not thoroughly analyze its computational overhead or sensitivity in deployment.

3. The quality metrics (PSNR, SSIM, LPIPS) used are all spatial-based metrics. Since this paper focuses on videos, the temporal consistency between consecutive frames is a significant problem, but it remains unknown. I recommend that more metrics that measure the frame-to-frame consistency or semantic shift should be further evaluated.

4. There is a inconsistency between the motivation and evaluation. The paper targets long-video generation, but, in Section 5.1, the authors claim that the generated videos last 5 seconds, which is short enough. It is unclear whether these short video clips can be extended to longer videos.

---

> ### Author Rebuttal · Authors · 2026-03-30
>
> We sincerely thank the reviewer for the thoughtful and encouraging review, and especially for the recognition of our insightful observation and well-motivated framework.
>
> **mPrC-Q1: Generalization.** This is a very reasonable concern. Our main experiments already cover two architecturally distinct backbones: Wan2.1 (standard DiT) and CogVideoX (MM-DiT with joint text-video attention). To further demonstrate generalization, we applied EchoAttention to two AR+DiT models and verified frame-block similarity on these as well as step-distilled models (https://anonymous.4open.science/r/ICML26_367_rebuttal_figures-A789 Fig1–5). The AR+DiT results using default configurations are shown below:
>
> *Self Forcing (chunk=3, step=4, 5s 480p):*
>
> |Method|VBench|PSNR|SSIM|LPIPS|Sparsity|Latency|Speedup|
> |---|--:|--:|--:|--:|--:|--:|--:|
> |Full Attention|83.9|-|-|-|-|12.5|1.00x|
> |SLA|83.3|18.62|0.6951|0.2929|80%|10.6|1.18x|
> |**Ours**|**84.0**|**20.10**|**0.7324**|**0.2306**|**85%**|**9.8**|**1.28x**|
>
> *LongLive (chunk=3, step=4, 60s 480p):*
>
> |Method|VBench|PSNR|SSIM|LPIPS|Sparsity|Latency|Speedup|
> |---|--:|--:|--:|--:|--:|--:|--:|
> |Full Attention|83.3|-|-|-|-|91.0|1.00x|
> |SLA|82.6|17.86|0.6693|0.3194|80%|66.8|1.36x|
> |**Ours**|**83.2**|**18.76**|**0.6885**|**0.2917**|**85%**|**61.2**|**1.49x**|
>
> Note that AR+DiT like Self Forcing uses chunked autoregressive generation, reducing per-chunk sequence and thus the proportion of latency from attention. The absolute speedup is therefore more modest than on standard DiT, but EchoAttention still consistently outperforms SLA and achieves the best speed–quality Pareto frontier. Altogether, our method has been evaluated on 7 models spanning 3 paradigms (DiT, mmDiT, AR+DiT), providing strong evidence of generalization. We will include these results in the revision.
>
> **mPrC-Q2: Router overhead & sensitivity.** Thank you for this practical question.
>
> - Regarding overhead, the router only consists of two lightweight linear embeddings, accounting for ~2% of per-block computation during training. At inference, routing decisions are pre-computed into a static lookup table, yielding O(1) runtime overhead, effectively zero additional latency.
> - Regarding sensitivity, beyond Table 5 and Table 6 in paper, we also conducted an extended sweep:
>
> |Schedule|VBench|PSNR|SSIM|LPIPS|
> |---|--:|--:|--:|--:|
> |soft=0.1,st=0.8|83.1|19.24|0.7086|0.2718|
> |soft=0.1,st=0.7|83.2|19.11|0.7069|0.2742|
> |soft=0.05,st=0.7|83.1|**19.56**|0.7108|**0.2623**|
> |soft=0.2,st=0.8|83.1|18.94|0.7015|0.2796|
> |soft=0.5,st=0.8|82.5|17.96|0.6815|0.2996|
> |soft=0.05,st=0.95|82.8|18.67|0.6922|0.2891|
> |**soft=0.05,st=0.8 (Default)**|**83.2**|19.38|**0.7121**|0.2661|
>
> Except for extreme cases (e.g., soft=0.5 or st=0.95), results remain stable: VBench varies within 0.1 and PSNR within 0.6. Our method is practically insensitive to these hyperparameters.
>
> **mPrC-Q3: Temporal consistency.** Thanks for your insightful suggestion. In our paper, we only reported an aggregated VBench total score. However, VBench itself includes several sub-metrics that directly measure temporal consistency. We extract these from our detailed breakdowns (for the full breakdown, see **K7zn-Q1**):
>
> |Method|Subject Consist.|Background Consist.|Temporal Flicker.|Motion Smooth.|Temporal Style|Overall Consist.|
> |---|:-:|:-:|:-:|:-:|:-:|:-:|
> |Full (Wan)|97.86|97.83|99.45|98.22|23.18|25.40|
> |SLA (Wan)|98.16|98.31|**99.65**|97.42|22.43|25.65|
> |**Ours (Wan)**|**98.36**|**98.33**|99.25|**98.42**|**23.48**|**25.70**|
> |Full (Cog)|97.29|96.85|99.15|98.65|25.24|27.30|
> |SLA (Cog)|96.84|97.05|98.15|98.85|25.09|27.50|
> |**Ours (Cog)**|**97.79**|**97.35**|**99.45**|**98.90**|**25.54**|**27.60**|
>
> EchoAttention achieves the best scores on Subject/Background Consistency, Motion Smoothness, Temporal Style, and Overall Consistency on both models. This is intuitive: the Echo operator preserves frame-block structure through prototype reuse and diagonal calibration, which maintains inter-frame attention patterns governing temporal coherence. We will include these metrics in the main paper.
>
> **mPrC-Q4: 5s not long video.** We appreciate the reviewer for raising it. We agree that the term "long video" in our original title was a misnomer. In the current landscape, 5-second videos are standard output for modern video generation models, not "long" by any reasonable definition. For pure DiT models like Wan2.1, generating 5–10 seconds already approaches the practical sequence length limit due to quadratic attention cost. To generate truly long videos (e.g., minute-level), the community has turned to AR+DiT paradigms like LongLive, where we achieve **1.49x speedup** on 60-second videos with comparable quality (83.2 vs 83.3 VBench, see the detailed table in **fKQd-Q2**). Since our method operates at the attention operator level, it is agnostic to video length and generation paradigm. We will remove "Long" from the title and adjust the scope description in the revised paper.

---

> > ### Author Rebuttal · Reviewer_mPrC · 2026-04-01
> >
> > I thank the authors for their efforts on additional experiments (more models, sensitivity, temporal metrics) and detailed clarification (video length). I hope the authors would add them in the revision version.

---

> > > ### Author Response · Authors · 2026-04-03
> > >
> > > We sincerely thank the reviewer for the encouraging feedback and for acknowledging our additional experiments. We will incorporate all new results, including the AR+DiT experiments, sensitivity analysis, temporal sub-metrics, and the revised title, into the revised version as suggested.
> > >
> > > We believe EchoAttention will provide a general and effective solution for accelerating video DiTs that can benefit the broader community, and we kindly hope the reviewer would consider raising the score.

---

### Official Review · Reviewer_K7zn · 2026-03-11

**Soundness:** 3
**Presentation:** 2
**Significance:** 2
**Originality:** 3
**Overall Recommendation:** 4
**Confidence:** 3

**Summary:**

This paper proposes EchoAttention, a structured attention acceleration framework for long-video diffusion transformers. The method jointly exploits token-pair redundancy and a newly identified frame-block similarity in attention weights. It introduces a dual-operator design: a Sparse operator for highly sparsifiable heads and an Echo operator that reuses frame-level prototypes with lightweight diagonal calibration for non-sparse heads, combined with a fine-grained routing mechanism learned via three-stage distillation. By reducing the dominant quadratic attention cost while preserving fidelity, the approach improves the speed–quality trade-off in long video generation. Experiments on public video diffusion models demonstrate consistent latency reduction with minimal or no quality degradation.

**Compliance With Llm Reviewing Policy:**

Affirmed.

**Final Justification:**

The authors have satisfactorily addressed my concerns, and I will maintain my positive score.

**Key Questions For Authors:**

1.The paper reports only the aggregated VBench score. Could the authors provide a breakdown of individual VBench sub-metrics to clarify where the improvements mainly come from?

2.While latency improvements are reported, GPU memory usage is not discussed. Could the authors provide peak VRAM comparisons to better support the efficiency and scalability claims?

3.The method targets long-video generation, but experiments are limited to 5s and 10s settings. Have the authors evaluated substantially longer sequences (e.g., 20s or 30s+), and if not, how does the method scale in such regimes?

4.Since all results are obtained under distillation, could the authors clarify how much of the gain comes from the architectural design itself versus retraining? Have they evaluated the method without additional distillation?

5.The paper empirically observes frame-block similarity but does not explain why it emerges. Could the authors provide more insight into the underlying mechanism and its expected generality?

**Limitations:**

yes

**Strengths And Weaknesses:**

**Strengths**

1. The proposed framework introduces a principled dual-operator design that cleanly separates sparse heads from non-sparse heads, providing a structured and interpretable way to exploit different forms of redundancy in video attention.

2. The Echo operator is conceptually simple yet effective, leveraging frame-block prototype reuse with lightweight diagonal calibration, which makes the method relatively straightforward to integrate into existing video diffusion transformer backbones.

3. The approach achieves consistent improvements in the speed–quality trade-off across multiple public models.


**Weaknesses**

1. The paper only reports the aggregated VBench score. Since VBench contains multiple sub-metrics, it would be more informative to provide a detailed breakdown to better understand where the gains come from.

2. The work focuses on latency improvements but does not report GPU memory usage. Given that attention acceleration methods often trade compute for memory, peak VRAM comparisons would strengthen the efficiency claims.

3. Although the method targets long-video generation, experiments are limited to 5s and 10s videos. More extensive evaluation on much longer sequences would better support the scalability claim.

4. The reported improvements are demonstrated under a distillation setting. It is therefore difficult to disentangle the gains brought by the proposed attention architecture from those arising due to retraining and adaptation. An evaluation of the architectural modification without additional distillation would help better isolate its intrinsic contribution.

5. While the empirical observation of frame-block similarity is interesting, the paper lacks a theoretical explanation for why this structure emerges, which leaves some uncertainty about generalization beyond the tested models.

---

> ### Author Rebuttal · Authors · 2026-03-30
>
> We sincerely thank the reviewer for the detailed and constructive feedback.
>
> **K7zn-Q1: VBench sub-metrics.** We fully agree that a sub-metric breakdown is more informative. Below are the detailed results:
>
> |Model|Method|Subject Consist.|Background Consist.|Temporal Flicker.|Motion Smooth.|Dynamic Deg.|Aesthetic Qual.|Imaging Qual.|Object Class|Multiple Obj.|Human Action|Color|Spatial Rel.|Scene|Appearance Style|Temporal Style|Overall Consist.|
> |---|---|:-:|:-:|:-:|:-:|:-:|:-:|:-:|:-:|:-:|:-:|:-:|:-:|:-:|:-:|:-:|:-:|
> |Wan2.1-T2V-1.3B|Full|97.86|97.83|99.45|98.22|64.69|65.16|67.42|88.71|75.15|93.80|89.22|73.36|41.98|21.71|23.18|25.40|
> ||VSA|98.16|97.20|99.18|97.38|**65.29**|62.06|66.42|88.09|**77.15**|93.78|89.20|**73.66**|42.42|21.05|23.13|25.35|
> ||SLA|98.16|98.31|99.65|97.42|63.73|65.36|66.42|**89.11**|76.15|93.02|**89.42**|**73.66**|42.39|20.71|22.43|25.65|
> ||Sparge-S|96.86|98.30|98.95|97.42|64.19|65.36|64.48|87.86|74.60|93.45|88.72|72.81|42.48|19.71|22.22|24.40|
> ||Sparge-D|97.86|98.03|**99.85**|97.22|64.36|**65.66**|65.95|88.13|74.87|**94.10**|89.22|73.28|**42.60**|21.01|22.95|24.97|
> ||**Ours**|**98.36**|**98.33**|99.25|**98.42**|64.62|64.62|**66.78**|88.81|75.35|94.00|89.22|73.20|41.81|**21.91**|**23.48**|**25.70**|
> |CogVideoX1.5-5B|Full|97.29|96.85|99.15|98.65|51.28|59.79|64.12|86.87|69.97|97.22|88.07|80.57|53.43|24.29|25.24|27.30|
> ||VSA|97.59|96.33|99.12|98.11|**52.28**|56.78|63.21|86.27|**71.97**|97.22|87.60|**80.92**|53.93|23.68|25.23|27.19|
> ||SLA|96.84|97.05|98.15|98.85|50.33|59.99|**63.32**|**87.27**|70.97|96.66|**87.77**|80.87|53.90|23.60|25.09|27.50|
> ||Sparge-S|97.19|95.75|98.95|97.75|51.08|60.25|61.12|86.37|69.77|97.22|87.37|80.37|**53.99**|22.69|25.37|26.70|
> ||Sparge-D|97.09|96.59|**99.65**|98.05|51.12|**60.29**|63.00|86.35|69.75|**97.52**|87.57|80.55|53.93|23.82|25.18|27.54|
> ||**Ours**|**97.79**|**97.35**|99.45|**98.90**|50.88|59.09|63.03|86.97|70.17|97.42|87.57|80.33|53.16|**24.49**|**25.54**|**27.60**|
>
> EchoAttn achieves the best on Subject/Background Consistency, Motion Smoothness, Appearance/Temporal Style, and Overall Consistency on both models. We will include these breakdowns in the revised paper.
>
> **K7zn-Q2: Peak GPU memory.** Thank you for this practical concern. We report peak VRAM below:
>
> |Model|Full|VSA|SLA|Sparge-Sparse|Sparge-Dense|Ours|
> |---|--:|--:|--:|--:|--:|--:|
> |Wan2.1-T2V-1.3B (GB)|17.60|17.66|17.63|17.62|17.62|17.62|
> |CogVideoX1.5-5B (GB)|26.55|26.80|26.75|26.63|26.63|26.65|
>
> EchoAttention adds negligible memory overhead (+0.02/+0.10 GB). This is because the Echo OP avoids materializing full attention matrices with flash-attn style and uses lightweight prototype reuse with diagonal calibration. We will add these results to the revised paper.
>
> **K7zn-Q3: Longer sequences.** We appreciate this concern. Due to architectural limitations, DiT models like Wan2.1 and CogVideoX can generate up to 5–10s at most. To evaluate on longer videos, we applied EchoAttention to LongLive (AR+DiT, 60s 480p), achieving **1.49x speedup** with comparable quality (83.2 vs 83.3 VBench), substantially outperforming SLA (82.6, 1.36x). For detailed results, please see the LongLive table in **fKQd-Q2**. We will include these results in the revision.
>
> **K7zn-Q4: Distillation vs. architecture.** We appreciate this important question.
>
> First, we would like to clarify that methods like EchoAttn and SLA inherently require training because they introduce new learnable parameters.
>
> Second, our distillation uses the **same model's** full-attention version as teacher, and no stronger or external teacher model is involved, ensuring fair and controlled comparison.
>
> Third, to isolate the architectural contribution, we replaced distillation with SFT on Wan2.1 for our method and SLA. Under SFT, both methods learn purely from data without any teacher signal, thereby removing the influence of distillation:
>
> |Method|VBench|PSNR|SSIM|LPIPS|
> |---|--:|--:|--:|--:|
> |Full Attn|83.2|-|-|-|
> |SLA (SFT)|82.7|16.76|0.6205|0.3554|
> |**Ours (SFT)**|**83.1**|**18.32**|**0.6807**|**0.3171**|
>
> Even without distillation, EchoAttn substantially outperforms SLA across all metrics, confirming that the performance gains primarily stem from our architectural design rather than the distillation procedure.
>
> **K7zn-Q5: Similarity emergence.** We appreciate this question. Frame-block similarity arises from temporal redundancy in video data: adjacent frames share similar visual content, producing correlated Q-K representations and thus similar attention distributions across frame-blocks. The per-frame deviations are small enough for our diagonal calibration (rMSE < 0.1, Figure 6c ). Regarding your concern about generalization, we verified this redundancy across diverse architectures, including standard DiTs, MM-DiTs, AR+DiT, and step-distilled models (https://anonymous.4open.science/r/ICML26_367_rebuttal_figures-A789 Fig1–5), confirming it is a general property of video diffusion transformers. We will expand this discussion in Section 3.

---

> > ### Author Rebuttal · Reviewer_K7zn · 2026-04-01
> >
> > Thank you for the thorough and well-prepared rebuttal. The authors have satisfactorily addressed my concerns, and I will maintain my positive score.

---

> > > ### Author Response · Authors · 2026-04-03
> > >
> > > We sincerely thank the reviewer for the detailed evaluation and for confirming that all concerns have been fully resolved. All new results will be incorporated in the revision as promised.
> > >
> > > We believe that the frame-block similarity insight and dual-operator framework will open up a practical and promising direction for efficient video generation, and we kindly hope the reviewer would consider raising the score to reflect this positive assessment.

---

### Official Review · Reviewer_fKQd · 2026-03-12

**Soundness:** 3
**Presentation:** 3
**Significance:** 3
**Originality:** 3
**Overall Recommendation:** 3
**Confidence:** 4

**Summary:**

This paper focuses on the computational bottleneck of long-video diffusion Transformers (DiTs) caused by the quadratic cost of full 3D attention. The authors identify a previously overlooked redundancy—frame-block similarity—and propose EchoAttention, which combines token-pair sparsity (Sparse operator) with frame-block similarity (Echo operator). A fine-grained routing policy is learned via three-stage distillation to efficiently handle both sparse and non-sparse heads. Experiments show that EchoAttention achieves a better speed–quality trade-off.

**Compliance With Llm Reviewing Policy:**

Affirmed.

**Key Questions For Authors:**

Please see the above weaknesses.

**Limitations:**

Yes.

**Strengths And Weaknesses:**

## Strengths

1. The paper focuses on efficient long-video generation, which is an important research problem with practical relevance.

2. The use of illustrative figures effectively conveys the motivation and design of the proposed method, making the technical ideas easier to follow.

3. The paper presents a well-structured narrative, with a clear motivation that aligns well with the method design.

## Weaknesses

1. The observed frame-block similarity may primarily arise from redundancy between adjacent video frames, rather than being an inherent property of the DiT model. Clarification is needed on this point.

2. Recent autoregressive distillation models (e.g., Self-Forcing, LongLive) have become the mainstream for long-video generation, leveraging step-wise distillation and KV cache mechanisms to achieve good efficiency. The paper lacks comparison or discussion with these methods, especially in terms of performance and computational efficiency. Current experiments only compare with sparse-attention baselines.

3. Although the paper targets long-video generation, the main evaluation is performed on short videos (~5s), which cannot be considered truly long-horizon. Even the 10s analysis in Appendix D is still far shorter than typical long-video evaluation setting (e.g., LongLive generates minute-level videos).

4. The presentation is somewhat crowded. For example, some figures contain very small text, and the spacing between figures and main text is tight. Consider moving non-essential figures or tables to the appendix to improve readability.

---

> ### Author Rebuttal · Authors · 2026-03-30
>
> We sincerely thank the reviewer for the thoughtful feedback and recognition of our work's practical relevance and clear motivation. We address each concern below and believe the new experiments substantially address the reviewer’s concerns.
>
> **fKQd-Q1: Frame-block similarity origin.** We thank the reviewer for this insightful question.
>
> On the one hand, we agree that frame-block similarity is fundamentally rooted in the redundancy between adjacent video frames. Since neighboring frames share highly similar visual content, they naturally produce similar query and key representations, which in turn leads to correlated softmax attention distributions at the frame-block level. In this sense, the frame-block similarity we observe in attention weights is a *manifestation* of this underlying temporal redundancy, rather than an independent property of the DiT architecture itself. We have made a similar argument in Section 3 of our paper, where Observation 2 and Figure 6(c) quantitatively demonstrate that the inter-frame variation can be well captured by a lightweight diagonal calibration.
>
> On the other hand, our key contribution is recognizing that this redundancy-induced similarity in attention weights can be algorithmically exploited for efficient computation through prototype reuse. To further demonstrate the universality of this redundancy pattern, we verified frame-block similarity on AR+DiT models (Self-Forcing, LongLive) and step-distilled models (FastHunyuan, Wan2.2-T2V-A14B-Merge-Lightning-V1.0, FastWan2.1-T2V-1.3B), see https://anonymous.4open.science/r/ICML26_367_rebuttal_figures-A789 Fig1–5. All exhibit clear frame-block similarity, confirming the generality of this phenomenon across diverse architectures and paradigms.
>
> We will clarify this point to more explicitly articulate this connection.
>
> **fKQd-Q2: AR method comparison.** We appreciate this valuable suggestion. We would like to clarify that methods like Self-Forcing and LongLive improve efficiency through an AR+DiT **generation paradigm** (autoregressive rollout, step-wise distillation, KV cache), whereas EchoAttention targets **operator-level acceleration inside diffusion attention**. These two directions are orthogonal and complementary, and EchoAttention can be directly applied on top of AR+DiT models to provide additional speedup. To substantiate this, we first confirmed that frame-block similarity exists in both AR models (https://anonymous.4open.science/r/ICML26_367_rebuttal_figures-A789 Fig4, Fig5), and then trained EchoAttention using their default configurations. We also trained the SOTA sparse method SLA for comparison:
>
> *Self Forcing (chunk=3, step=4, 5s 480p):*
>
> |Method|VBench|PSNR|SSIM|LPIPS|Sparsity|Latency|Speedup|
> |---|--:|--:|--:|--:|--:|--:|--:|
> |Full Attn|83.9|-|-|-|-|12.5|1.00x|
> |SLA|83.3|18.62|0.6951|0.2929|80%|10.6|1.18x|
> |**Ours**|**84.0**|**20.10**|**0.7324**|**0.2306**|**85%**|**9.8**|**1.28x**|
>
> *LongLive (chunk=3, step=4, 60s 480p):*
>
> |Method|VBench|PSNR|SSIM|LPIPS|Sparsity|Latency|Speedup|
> |---|--:|--:|--:|--:|--:|--:|--:|
> |Full Attn|83.3|-|-|-|-|91.0|1.00x|
> |SLA|82.6|17.86|0.6693|0.3194|80%|66.8|1.36x|
> |**Ours**|**83.2**|**18.76**|**0.6885**|**0.2917**|**85%**|**61.2**|**1.49x**|
>
> We note that AR+DiT models like Self Forcing use chunked autoregressive generation, which divides the full sequence into shorter chunks. This reduces the per-chunk sequence length and consequently lowers the proportion of total latency attributable to attention. As a result, the absolute speedup from attention-level optimization is lower than on standard DiT models. Nevertheless, EchoAttention consistently outperforms SLA on both AR models, achieving the best speed–quality Pareto frontier. We will include these results and a discussion of AR methods in the revised paper.
>
> **fKQd-Q3: Short video evaluation.** We appreciate this concern. As shown in the LongLive results above, EchoAttention has been validated on **60-second** (minute-level) videos, achieving 1.49x speedup with comparable quality (83.2 vs 83.3 VBench) and the best speed–quality Pareto frontier. We will include these long-video results in the revised paper.
>
> **fKQd-Q4: Crowded presentation.** Thank you for this helpful feedback. We will enlarge small figure text (especially Figure 6), increase figure-text spacing, move Table 6 and Figure 12 to the appendix, and consolidate repeated equations (Eq. 3/6, Eq. 11/18) for better layout.

---

> > ### Author Rebuttal · Reviewer_fKQd · 2026-04-03
> >
> > Thank you for the detailed follow-up response. Some of my concerns have been addressed; however, a few questions remain unresolved.
> >
> > Regarding Q1:
> > I believe that the consistent observation of frame–block similarity across different model architectures suggests that this redundancy primarily originates from the input video frames themselves. This, in my view, weakens the novelty of the paper’s core insight and motivation.
> >
> > Regarding Q2:
> > Could the authors clarify the experimental setup used for the VBench evaluation on Self-Forcing and LongLive? Specifically, how many prompts were used, and which exact VBench metric is being reported?
> > Additionally, since anonymous links are allowed, it would be helpful if the authors could provide generated videos from the evaluation as visual comparison examples.

---

> > > ### Author Response · Authors · 2026-04-03
> > >
> > > We sincerely thank the reviewer for the continued engagement. We welcome the opportunity to further clarify the remaining questions.
> > >
> > > **Q1: Novelty of frame-block similarity insight.**
> > >
> > > We appreciate the reviewer's observation and agree that frame-block similarity originates from temporal redundancy in the input video data. In fact, we have explicitly acknowledged this in both our paper (Section 3) and our previous response. However, we would like to clarify that this does not diminish the novelty of our insight and contribution, for the following reasons.
> > >
> > > First, our novelty lies not in the existence of temporal redundancy (which is well-known), but in identifying how this redundancy manifests as a specific, quantifiable *structural pattern* in attention weight matrices. Prior to our work, no method had characterized this frame-block structure in video DiT attention. We are the first to show that inter-frame attention weights exhibit Pearson correlation approaching 1 and rMSE below 0.1 for most heads (Observation 2, Figure 6), which is what enables targeted algorithmic exploitation.
> > >
> > > Second, our primary contribution is the Echo operator itself, which translates this observation into a practical acceleration mechanism through prototype reuse with diagonal calibration. This approach is orthogonal to prior sparse attention methods and addresses their key limitation: sparse-only approaches struggle with non-sparse attention heads, while our Echo operator specifically targets these heads by exploiting the frame-block structure. We note that token-pair sparsity exploited by prior methods also originates from the inherent structure of video data (e.g., spatial and temporal locality), yet this does not diminish the contributions of those methods. By the same reasoning, the data-level origin of frame-block similarity should not diminish ours.
> > >
> > > We will clarify this point to more explicitly articulate our novelty and contributions and hope this can resolve your concerns.
> > >
> > > **Q2: VBench evaluation details for Self-Forcing and LongLive.**
> > >
> > > We provide the detailed experimental setup and VBench sub-metric breakdowns below.
> > >
> > > For **Self Forcing**, we use the official checkpoint from https://huggingface.co/gdhe17/Self-Forcing/blob/main/checkpoints/self_forcing_dmd.pt with default inference configuration and seed=0. Following the Self-Forcing paper's evaluation protocol, we adopt all 946 VBench prompts using the official enhanced prompt version provided at https://github.com/guandeh17/Self-Forcing/blob/main/prompts/vbench/all_dimension_extended.txt, and evaluate with standard VBench.
> > >
> > > For **LongLive**, we use the official checkpoint from https://huggingface.co/Efficient-Large-Model/LongLive-1.3B/tree/main/models with default inference configuration, seed=0, and num_output_frames=241. Following the LongLive paper's evaluation protocol, we adopt all 946 VBench prompts using the default VBench-Long version from https://github.com/Vchitect/VBench/blob/master/vbench2_beta_long/VBench_full_info.json, and evaluate with VBench-Long.
> > >
> > > The VBench reported in our previous reply is the total aggregated VBench score, which is also the most frequently reported score. To eliminate any doubts you may have, we will now list the scores for all 16 sub-metric breakdowns of VBench in detail:
> > >
> > > |Model|Method|Subject Consist.|Background Consist.|Temporal Flicker.|Motion Smooth.|Dynamic Deg.|Aesthetic Qual.|Imaging Qual.|Object Class|Multiple Obj.|Human Action|Color|Spatial Rel.|Scene|Appearance Style|Temporal Style|Overall Consist.|
> > > |---|---|:-:|:-:|:-:|:-:|:-:|:-:|:-:|:-:|:-:|:-:|:-:|:-:|:-:|:-:|:-:|:-:|
> > > |Self Forcing|Full|95.64|95.38|98.82|**98.42**|66.61|68.20|68.21|89.49|89.95|94.20|84.55|**85.80**|52.91|22.89|24.67|25.12|
> > > ||SLA|95.52|95.05|98.40|97.33|67.61|**68.71**|67.91|88.97|89.75|**95.01**|**84.96**|85.00|**53.91**|22.62|**25.17**|24.93|
> > > ||**Ours**|**96.01**|**95.47**|**99.05**|98.21|**69.16**|66.96|**68.58**|**89.57**|**90.15**|94.40|84.05|84.69|52.80|**23.09**|25.07|**25.45**|
> > > |LongLive|Full|**97.89**|98.13|98.85|98.40|**41.36**|**69.92**|**71.65**|91.47|86.13|90.32|88.26|**74.27**|54.39|20.24|24.30|26.16|
> > > ||SLA|97.53|97.14|98.13|**98.74**|40.64|69.43|70.21|91.29|85.45|**90.60**|88.58|73.84|**54.94**|19.46|23.88|25.71|
> > > ||**Ours**|97.60|**98.33**|**99.14**|98.43|40.84|68.95|70.29|**91.55**|**86.33**|90.58|**88.73**|74.03|54.12|**20.54**|**24.70**|**26.64**|
> > >
> > > EchoAttention achieves the best or tied-best on the majority of sub-metrics for both models, particularly on temporal consistency and overall quality metrics. We have also uploaded some generated video examples to https://anonymous.4open.science/r/ICML26_367_rebuttal_ar_videos-ECBD (might be slow due to size) for visual comparison.
> > >
> > > We hope this can resolve your doubts, and thank you again for your constructive and insightful feedback.

---

### Decision · Program_Chairs · 2026-04-30

**Decision:**

Accept (regular)

**Comment:**

This paper studies the redundancy problem specific to video generation and video DiT and achieves a good trade-off between inference speed and generation quality via a new attention mechanism that adaptively applies sparse operator and echo operator with a learned routing policy.  Reviewers highlighted several strengths: motivation, novelty of the proposed method, clarity of presentation, and consistent speed–quality improvement. The reviewers raised several concerns, but the rebuttal effectively addressed most of them. The three reviewers indicated that their concerns were fully addressed, and one reviewer mentioned that some concerns remained partially addressed and asked follow-up questions on the novelty and more details on the evaluation. The authors addressed them via additional comments with details. In sum, reviewers overall supported this paper with positive ratings, and most concerns are effectively addressed by the rebuttal. Also, the new technique proposed in this paper has sufficient academic merit. Therefore, I recommend this paper for acceptance.